# Type 1 Diabetes Mellitus and Vitamin D

**DOI:** 10.3390/ijms26104593

**Published:** 2025-05-11

**Authors:** Teodoro Durá-Travé, Fidel Gallinas-Victoriano

**Affiliations:** 1Department of Pediatrics, School of Medicine, University of Navarra, Avenue Irunlarrea, 4, 31008 Pamplona, Spain; 2Navarrabiomed (Biomedical Research Center), 31008 Pamplona, Spain; fivictoriano@hotmail.com; 3Department of Pediatrics, Navarra University Hospital, 31008 Pamplona, Spain

**Keywords:** islet autoimmunity, insulitis, Type 1 diabetes mellitus, vitamin D status, vitamin D deficiency, vitamin D supplementation, cholecalciferol, calcitriol, calcidiol, alfacalcidol, ergocalciferol

## Abstract

Type 1 diabetes mellitus (T1DM) is a multifactorial disease in which environmental factors and genetic predisposition interact to induce an autoimmune response against pancreatic β-cells. Vitamin D promotes immune tolerance through immunomodulatory and anti-inflammatory functions. The aim of this study is to provide a narrative review about the association between vitamin D status in the pathogenesis of T1DM and the role of vitamin D supplementation in the prevention and treatment of T1DM. Although vitamin D deficiency is more prevalent in children/adolescents with new-onset T1DM than in healthy individuals, there does not appear to be an association between vitamin D status before diagnosis and the onset of T1DMD later in life. The results of vitamin D as adjuvant therapy have, at best, a positive short-term effect in newly diagnosed T1DM patients. Intervention studies have been conducted in the clinical phase of T1DM, but it would be desirable to do so in the early stages of the autoimmune process (pre-diabetes).

## 1. Introduction

Type 1 diabetes mellitus (T1DM) is one of the most common chronic autoimmune diseases that tends to occur in children and adolescents, although it can affect people of any age. T1DM results from autoimmune destruction of the β-cells in the pancreatic islets of Langerhans, ultimately leading to absolute insulin deficiency and lifelong dependence on exogenous insulin [1]. Interestingly, the incidence of T1DM has increased significantly in recent decades [2]. At the same time, the global incidence of vitamin D deficiency has increased in all age groups, including children and adolescents [3]. It should be noted that the incidence of T1DM is higher in high-latitude regions (e.g., Canada and Scandinavian countries) where there is less exposure to sunlight and, consequently, a higher incidence of vitamin D deficiency, leading to the hypothesis that vitamin D may play a role in the T1DM process [4,5]. In addition, a recent meta-analysis found a positive association between the latitude of the patient’s residence and the risk of T1DM. This also suggests that people living at high latitudes may be predisposed to T1DM, whereas people living near the equator synthesize enough vitamin D due to the strong solar ultraviolet B radiation available [6]. In addition, a number of observational studies have shown that children with new-onset T1DM tend to have significantly lower vitamin D levels than healthy controls children [7,8,9,10,11,12,13].

The physiological role of vitamin D goes far beyond the regulation of calcium homeostasis and bone metabolism, as it exerts a wide variety of extra-skeletal effects. Indeed, vitamin D is now considered a pleiotropic hormone, exerting its effects through both genomic and non-genomic actions. Most of the effects of vitamin D are mediated by its interaction with a nuclear transcription factor or vitamin D receptor (VDR) and subsequent binding to specific DNA sequences, regulating the expression of a large number of target genes involved in several physiological processes, including cellular proliferation and differentiation, as well as anti-inflammatory and immunomodulatory activities (genomic pathway) [5,11,12].

Because of the widespread distribution of VDRs throughout the human body, including immune cells (antigen-presenting cells and activated T and B lymphocytes) and pancreatic β cells, several experimental and epidemiological studies support the ability of vitamin D to prevent the pathogenesis of T1DM [14]. This suggests that vitamin D deficiency may be an important environmental factor in the development of the disease. The beneficial effects of vitamin D in T1DM would be based on its functional versatility in various immune populations, such that vitamin D could improve glucose homeostasis by preserving β-cell mass, reducing inflammation, and decreasing autoimmunity. In fact, over the last decade, numerous studies have shown associations between vitamin D deficiency and the risk of developing autoimmune diseases, including T1DM [6,15,16,17,18,19,20].

The aim of this review is to produce a comprehensive literature review (narrative review) on (a) research progress on the possible function of vitamin D status as an environmental risk factor in the pathogenesis of T1DM and (b) the assessment of the potential role of vitamin D in the prevention and treatment of T1DM. This review is based on an electronic search of the PubMed database of the US National Library of Medicine for literature published between January 2011 and December 2024, conducted by two independent researchers. The following specific keywords (Medical Subject Headings) were used alone or in combination for the search: “vitamin D” and “Type 1 diabetes mellitus”.

## 2. Vitamin D Synthesis and Metabolism

Vitamin D is known as the “sunshine hormone”. In fact, it is a steroid hormone that is mainly obtained (80–90%) through exposure to sunlight (vitamin D3) and, to a lesser extent, from food (vitamin D2). When exposed to solar ultraviolet B radiation, 7-dehydrocholesterol present in the human skin is converted to vitamin D3. Vitamin D (including vitamin D2 or D3) in the circulation is bound to vitamin D binding protein (VDBP), which transports it to the liver. There, vitamin D is converted by vitamin D 25-hydroxylase (encoded by the CYP2R1) to 25-hydroxyvitamin D [25(OH)D] or calcidiol (used as a biomarker for vitamin D status). 25(OH)D then circulates bound to VDBP and reaches the kidneys, where it is activated by 25(OH)D-1αhydroxylase (encoded by the CYP27B1) to 1,25-dihydroxyvitamin D [1,25(OH)2D] or calcitriol, which is the biologically active form of vitamin D.

According to the US Endocrine Society’s guidelines, calcidiol levels are considered the best indicator of organic vitamin D content, given its long half-life (two to three weeks). Vitamin D deficiency is defined as calcidiol lower than 20 ng/mL (<50 nmol/L). Vitamin D insufficiency is when calcidiol levels fluctuate between 20 and 29 ng/mL, and vitamin D sufficiency is when calcidiol levels reach or overtake 30 ng/mL (>75 nmol/L). That is, optimal vitamin D levels range between 30 and 50 ng/mL (75–125 nmol/L), and maximum safe levels go up to 100 ng/mL (250 nmol/L) [21].

For the genomic pathways (Figure 1), vitamin D binds to the vitamin D receptor (VDR), which acts as a transcription factor by forming a heterodimer with the retinoid X receptor (RXR). The VDR/RXR complex then translocates to the nucleus and binds to specific nucleotide sequences in DNA (also known as VDRE, vitamin D response elements) to modulate the expression of a substantial number of target genes (about 5–10% of the total human genome). In addition, vitamin D induces rapid non-genomic cellular responses by binding to a membrane receptor, called the membrane-associated rapid response steroid (MARRS), to regulate several intracellular processes (calcium transport, mitochondrial function, ion channel activity, etc.) [5,11,12,22,23].

## 3. Pathogenesis and Natural History of Type 1 Diabetes Mellitus

T1DM is a complex multifactorial disease in which environmental factors and genetic predisposition interact to promote the induction of an autoimmune response against β-cells [24]. Both humoral and cellular immune responses are involved in the pathogenesis of T1DM. The natural history of T1DM shows that its clinical diagnosis occurs several years after the onset of the autoimmune process of β-cell damage. In fact, by the time T1DM is diagnosed, approximately 70–80% of the β-cell mass has been destroyed [25].

Genetic predisposition to β-cell autoimmunity and, subsequently, T1DM occurs mainly in individuals with specific human leukocyte antigen (HLA) class II haplotypes involved in antigen presentation: HLA-DR3-DQ2, HLA-DR4-DQ8, or both. However, other genes may also be involved, which are either related to autoimmunity (CTLA4, PTPN22, IL2RA, etc.) or β-cells (INS, GLIS3, CTSH, etc.) [26,27,28]. The increase in the prevalence of T1DM in recent decades cannot be attributed to genetic factors alone, as environmental factors most likely play a role in triggering islet autoimmunity. In addition, evidence from the incomplete concordance of diabetes incidence in monozygotic twins suggests that environmental factors also play critical a role in T1DM pathogenesis. In other words, the selective destruction of β-cells could be caused by an interaction between risk genes and environmental factors.

To date, viral infections have been considered the most important environmental factors in initiating the process of autoimmune destruction of β-cells. There is no specific “diabetes virus”, but several agents associated with diabetes have been described, including rubella, mumps, and measles viruses, varicella-zoster virus, enterovirus, adenovirus, rotavirus, coxsackievirus, cytomegalovirus, parvovirus B19, Epstein–Barr virus, human endogenous retroviruses, and also SARS-CoV-2 [29]. Viruses can damage pancreatic β-cells either through direct cytolysis or by inducing an autoimmune response against β-cells. It is now believed that viral induction of an autoimmune response may be the result of a phenomenon of molecular mimicry, as some similarities have been described between the amino acid sequence of some β-cell molecules and proteins derived from several viruses (adenovirus, Epstein–Barr virus, coxsackievirus, cytomegalovirus, etc.). This means that the similarity of particle fragments derived from viral proteins to β-cell antigens may contribute to the activation of autoreactive T cells and the initiation of the autoimmunity process leading to the destruction of the pancreatic islets. For virions to be destroyed in the host, the viral peptides must be presented by antigen-presenting cells (such as dendritic cells and macrophages). If the presented antigens are molecularly similar to those of the organism, autoreactive B and T lymphocytes are produced. As a result, these cells may cross-react with virions and self-antigens on the surface of pancreatic cells (molecular mimicry model). However, it is extremely difficult to prove a cause–effect relationship between viral infections and the development of T1DM, as the period between viral exposure and the onset of clinical symptoms of T1DM is often very long.

In recent years, in addition to viral infections, several epidemiological studies have described new environmental factors, such as nutrition in the first months of life, gut microbiota, and climatic conditions, including higher latitudes and reduced sunlight exposure and/or vitamin D deficiency, which may play a role in triggering islet autoimmunity and T1DM in subjects at high genetic risk. While breastfeeding has a protective role against autoimmunity, studies show conflicting results on the possible impact of cow’s milk, gluten, or omega-3 fatty acid intake in children on the risk of T1DM [30,31].

A complex correlation between gut microbiota, the immune system, and intestinal permeability has been identified, although it has not been fully unraveled. Several cross-sectional studies have found large significant differences in microbiota composition between subjects with T1DM or islet autoimmunity and healthy controls. Hypothetically, intestinal dysbiosis would lead to dysregulation of the immune response, including both the innate and adaptive immune systems, and/or intestinal permeability, ultimately leading to beta cell destruction and the onset of T1DM in genetically susceptible individuals. Increased intestinal permeability would allow antigens and pathogens to cross the intestinal barrier, which could trigger or exacerbate immune responses against pancreatic beta cells, contributing to the etiopathogenesis of T1DM. However, the exact role of the intestinal microbiota in the pathogenesis of T1DM remains controversial [32,33,34,35]. All of these factors could alter gene expression through epigenetic mechanisms (particularly DNA methylation, histone modifications, or long non-coding RNA), thereby inducing an aberrant immune response and progressive β-cell destruction (Figure 2), and thus may be involved in the pathogenesis of T1DM [28,36,37].

The core of the autoimmune process in T1DM is a breakdown of immune self-tolerance to pancreatic β-cell autoantigens, resulting in their destruction by infiltration of the pancreatic islets by monocytes/macrophages, natural killer (NK) cells, helper (CD4+) and cytotoxic T cells (CD8+), and plasma cells, known as autoimmune insulitis. β-cells are destroyed through apoptosis and necrosis (note: immune tolerance is a physiologic state where the immune system does not react against the body’s own cells and tissues, preventing an immune attack against self-antigens. This is an active state (not a simple absence of response) endowed with specificity and memory. Autoimmune diseases arise when this tolerance is disrupted, leading to the immune system attacking the body’s own tissues).

First, in children at high genetic risk of T1DM, viral infections or other environmental factors cause β-cells’ apoptosis, resulting in the release of β-cell antigens. Self-antigens induce the activation of antigen-presenting cells (APCs). Activated APCs, mainly dendritic cells (DCs) and NKs, present self-antigens to naïve CD4+ T cells in the pancreatic lymph nodes and promote their activation. Subsequently, CD4+ T cells expand and differentiate into several effector subpopulations, including Th1, Th2, Th17, and Treg cells, with IL-12 produced by APCs mainly inducing Th1 differentiation. Specifically, Th1 cells (autoreactive T cells) infiltrate pancreatic islets and release pro-inflammatory interleukins (IL), such as IL-2, tumor necrosis factor α (TNF-α) and interferon γ (IFN-γ), which induce the migration of macrophages and NK cells to pancreatic islets (insulitis) and enhance the destruction of pancreatic β-cells through apoptosis/necrosis. In addition, Th1 cells activate CD8+ T cells (cytotoxic T cells) via IL-2 and IFN-γ, which also induce pancreatic β-cell apoptosis/necrosis. Although increased Th1 cell activity, also known as the Th1 profile, and, consequently, Th1/Th2 cell imbalance have been considered the main contributors to the development of T1DM, other authors suggest that an increased Th17/Treg ratio could also play a key role in the pathogenesis of T1DM.

In addition to autoreactive T cells against β-cell autoantigens, which are considered an important factor in cell damage, a humoral response is also involved in the mechanism of the autoimmune process and the destruction of pancreatic islets. Th1 recruitment is associated with the activation and expansion of B lymphocytes and their subsequent differentiation into autoantibody-producing plasma cells, which further contribute to the inflammatory process and tissue destruction. A higher proportion of plasma cells in insulitis has been associated with faster β-cell decline [38].

In fact, the clinical symptoms of T1DM are preceded by a long period of “pre-diabetes”, characterized by specific histological features (autoimmune insulitis) and the presence of serum islet autoantibodies, including anti-insulin antibodies (IAAs), antibodies against insulin-producing islet cells (ICAs) and/or antibodies to glutamic acid decarboxylase (GAD65), antibodies to insulinoma-associated antigen-2 (anti-IA2), and antibodies to zinc transporter 8 (ZnT8), although patients remain normoglycemic and asymptomatic (stage 1). Subsequently, patients retain islet autoantibody positivity and remain asymptomatic but exhibit dysglycemia, as evidenced by an abnormal glucose tolerance test (stage 2). The persistent presence of at least two types of β-cell autoantibodies, which are considered predictive biomarkers of irreversible islet autoimmunity [39], lead to the onset of clinical T1DM (symptomatic disease). The first clinical symptoms of the disease (polyuria, polydipsia, fatigue, weight loss, diabetic ketoacidosis, etc.) usually appear several years after the onset of the autoimmune process, when most of the pancreatic β-cells have been destroyed (stage 3) and T1DM is definitively established [27,40,41].

Nevertheless, a few weeks after clinical onset of the disease and the initiation of insulin therapy, about half of all patients with T1D experience a transient and partial spontaneous remission or “honeymoon phase” [42]. During this phase, the remaining β-cells are still able to produce sufficient insulin, leading to a significant reduction in exogenous insulin requirements. Although the pathogenesis of the remission phase remains unknown, it is of remarkable clinical importance, as it can be used to investigate the potential efficacy of different therapeutic agents to halt or slow down the autoimmune process and disease progression in T1DM [43].

Reactive oxygen species (ROS) include several oxygen-containing free radicals, such as superoxide anion, hydroxyl radical, and hydrogen peroxide. Excessive amounts of ROS may cause deleterious oxidative damage to biomolecules (DNA, proteins, and lipids), consequently leading to cell death. An imbalance between the production and accumulation of ROS and enzymatic antioxidant systems, including glutathione peroxidase, superoxide dismutase, and catalase, is known as “oxidative stress”. The main source of free radicals responsible for oxidative stress is mitochondrial respiration, but they can originate from exogenous sources, such as aging, inflammation, radiation, and toxic chemicals. That is, chronic inflammatory processes can aggravate oxidative stress and enhance ROS generation [44,45]. Although hyperglycemia is a key factor in oxidative stress, the interaction of autoimmune and inflammatory pathways in T1DM, independent of glucose levels, could contribute to the overproduction of ROS. These ROS can cause oxidative damage to cellular structures, leading to β-cell death and disease progression. In fact, mitochondria-derived free radicals have been shown to contribute to the immune-mediated β-cell destruction process, either through the induction of toxicity by cytokines (IL-1, TNF-α, and IFN-γ) or due to the low expression of antioxidant enzymes in islets [46,47,48].

## 4. Immunomodulatory Effects of Vitamin D in Autoimmune Diseases

The immunomodulatory effect of vitamin D is based on a genomic response and its ability to modify gene transcription. The most important role of vitamin D in autoimmune diseases, including T1DM, is its ability to induce immune tolerance as well as anti-inflammatory effects. A brief summary of the immunomodulatory effects of vitamin D on immune cells is presented in Table 1.

Vitamin D is involved in the regulation of both innate and adaptive immunity. Effects of vitamin D on both NKs, macrophages, and DCs include inhibition of inflammatory cytokine release (IL-1, IL-2, IL-6, IL-8, IL-12, and IFN-γ) and increased production of anti-inflammatory cytokines, such as IL-4 and IL-10. It also affects the differentiation of DCs, resulting in the preservation of immature DCs (tolerogenic phenotype) and, consequently, a reduction in the number of antigen-presenting cells and activation of naïve CD4+ T cells, thus contributing to the induction of a tolerogenic state. Inhibition of DCs’ differentiation and maturation therefore induces T-cell anergy (unresponsiveness), which is particularly important in the context of an autoimmune process.

Vitamin D promotes the differentiation of T-helper lymphocytes from a Th1 to a Th2 phenotype. This change implies inhibition of inflammatory cytokine production (IL-2, IFN-γ, and TNF-α) and an increased production of anti-inflammatory cytokines (IL-4, IL-5, and IL-10). Furthermore, vitamin D affects differentiation to the Th17 phenotype, leading to a decrease in the production of inflammatory cytokines (IL-17 and IL-21), and facilitates the induction of T reg cells with increased production of anti-inflammatory cytokines (IL-10 and TGF-β). In addition, Treg cells are also capable of suppressing the proliferation of CD8+ (cytotoxic lymphocytes) and antigen-presenting cells. Thus, vitamin D could modulate cell-mediated immune responses and regulate the inflammatory activity of T cells, an important role in the prevention of autoimmune responses.

With respect to B-lymphocyte regulation, vitamin D impairs B cell activation and proliferation, plasma cell differentiation, memory B cell formation, and autoantibody production. These effects of vitamin D on B-lymphocyte homeostasis may be clinically relevant in T1DM, as autoreactive antibodies are involved in the pathophysiology of autoimmunity.

In summary, the immunomodulatory effect of vitamin D is characterized by the induction of immune tolerance and T cell anergy, impairment of B cell activity and autoantibody production, and reduction of the inflammatory response. Therefore, it can be suggested that vitamin D could play an important therapeutic role in reducing the risk of the autoimmune process of T1DM [5,40,41,49,50,51,52,53,54,55].

## 5. Vitamin D Status and the Risk of Type 1 Diabetes Mellitus

A multicenter study (the POInT study) was recently conducted with a large sample of infants and children aged 4–7 months to 3 years (longitudinal design) with a high genetic risk of developing β-cell autoantibodies, as defined by HLA genotype or family history of T1DM, in five European countries (Belgium, England, Germany, Poland, and Sweden). Multivariate logistic regression analysis concluded that vitamin D deficiency is common and persistent in infants and children with a genetic predisposition to developing T1DM [15]. A systematic review and meta-analysis was recently conducted that included a total of 45 studies of acceptable quality [56] with statistical information on vitamin D deficiency in children and adolescents with T1DM. These studies included 6995 participants from 25 countries in Africa (Egypt and Tunisia), Oceania (Australia), Europe (the United Kingdom, Spain, Italy, Poland, Slovakia, Switzerland, Ukraine, and Germany), North America (the USA and Canada), and Asia (Turkey, Korea, Iran, India, China, the Kingdom of Saudi Arabia, Indonesia, Kuwait, Bangladesh, and Iraq). The results of this meta-analysis showed that the proportion of vitamin D deficiency among children/adolescents with T1DM was 45%. Several authors who have measured serum vitamin D levels in newly diagnosed children have observed significantly lower serum vitamin D levels at the onset of T1DM compared to controls [7,8,10,13,17]. In addition, serum vitamin D levels were significantly lower in patients admitted with diabetic ketoacidosis than in patients without ketoacidosis [7,9]. Although numerous researchers have suggested an association between vitamin D deficiency and T1DM, it is not entirely clear whether vitamin D insufficiency is a trigger for T1DM or a consequence of the disease.

There is now a body of literature suggesting that vitamin D status may be an important environmental risk factor in the pathogenesis of T1DM, rather than a consequence of pathophysiological changes resulting from the disease. Numerous authors have investigated whether polymorphisms of genes involved in vitamin D metabolism, especially those encoding vitamin D hydroxylases and VDBP, may influence the risk of islet autoimmunity and T1DM. Some observational studies have reported that various polymorphisms in CYP2R1 (the gene encoding vitamin D 25-hydroxylase), CYP27B1 (the gene encoding vitamin D 1α-hydroxylase), and the VDBP gene were significantly associated with an increased risk of T1DM [40]. In contrast, a recent Mendelian randomization analysis involving 9356 cases (from Canada, the United Kingdom, and the United States) failed to demonstrate an association between any polymorphism in these genes and the risk of T1DM [57]. Similarly, a recent systematic review and meta-analysis that included studies from different geographical locations did not confirm the hypothesis that polymorphisms in these vitamin-D-related genes might be associated with an increased risk of T1DM [58]. Some studies have shown an association between VDBP genetic polymorphisms and the risk of T1DM, but these results have not been subsequently confirmed.

However, a potential role of VDR gene polymorphisms in T1DM seems more suggestive. There are four known VDR polymorphisms that have been extensively studied for their potential role in T1DM: *ApaI*, *BsmI*, *TaqI*, and *FokI*. For example, the Environmental Determinants of Diabetes in the Young (TEDDY) study is a large prospective study of children at increased T1DM risk, as defined by HLA genotype or family history of T1DM, conducted at centers in the USA and Europe. The study found that higher serum vitamin D concentration was associated with a decreased risk of islet autoimmunity, but only in those with VDR gene *Apal* polymorphism. That is, a higher concentration of circulating vitamin D in combination with VDR genotype (Apal) may decrease the risk of developing autoimmune insulitis, suggesting that the underlying mechanism involves vitamin D action [18]. In addition, the results of a meta-analysis conducted in Asian populations (China, Japan, India, Iran, and Turkey) suggested that the *BsmI* polymorphism might be a risk factor for susceptibility to T1DM in the East Asian population and that the *FokI* polymorphism was associated with an increased risk of T1DM in the West Asian population. However, the authors reported that the statistical power was not sufficient due to the limited number of included articles and concluded that further studies are essential to confirm their findings [59]. Furthermore, a study (case-control design) was recently conducted in three major hospitals in Kuwait (Adan, Farwania, and Mubarak Al-Kabeese), and the genotypes of four VDR gene polymorphisms were determined in 253 children with newly diagnosed T1DM. The results of that study show that VDR gene *FokI* and *TaqI* polymorphisms are associated with susceptibility to T1DM and thus contribute significantly to genetic predisposition to T1DM [19]. Finally, a prospective cohort study of 101 children with newly diagnosed T1DM demonstrated that adequate vitamin status (≥30 ng/mL) together with the *FokI* and *TaqI* polymorphisms of the VDR gene could lead to greater preservation of residual β-cell mass and function [60]. However, although VDR gene polymorphisms have recently been associated with susceptibility to various autoimmune diseases, there is no comprehensive meta-analysis of VDR gene polymorphisms and the risk of T1DM, and the existing literature is relatively inconsistent [61].

In 2012, the first study (case-control design) reporting an association between low vitamin D levels during pregnancy and an increased risk of T1DM in offspring was conducted in a cohort of 35,940 pregnant women in Norway. Because of the long follow-up period (15 years), virtually all children in the original maternal cohort who developed T1DM in childhood could be identified. The study showed that the mothers of children who developed T1DM before the age of 15 had significantly lower serum vitamin D levels during the last trimester of pregnancy than the mothers of children who did not develop the disease [62]. In contrast, a case-control study of 383 Finnish pregnant women (Finnish Maternity Cohort) found no difference in serum vitamin D levels during the first trimester of pregnancy between mothers of children who subsequently developed T1DM and mothers of non-diabetic children of the same age (0–7 years) [63]. A report based on the All Babies In Southeast Sweden (ABIS) study, which used questionnaire data from 16,339 mothers, found no significant association between vitamin D intake during pregnancy and the risk of T1DM in children aged 14–16 years [64]. Similarly, the aforementioned TEDDY study found no association between maternal vitamin D supplementation during pregnancy and an increased risk of islet autoimmunity [65]. Finally, a meta-analysis of relevant observational studies found insufficient evidence for an association between maternal vitamin D intake and risk of T1DM in offspring [66]. That is, given the limited and inconsistent evidence at present, large randomized controlled trials with long-term follow-up are needed to clarify whether prenatal vitamin D exposure modifies the risk of T1DM later in life. It should be noted that vitamin D supplementation in early childhood appears to play a more relevant role than prenatal vitamin D exposure in determining the risk of T1DM. In fact, several meta-analyses of observational studies suggest that there is evidence that vitamin D supplementation in infancy may offer protection against the development of T1DM [66,67].

Results from long-term follow-up studies in children suggest no association between pre-diagnosis vitamin D status and the occurrence of T1DM later in life. For example, in a cross-sectional study of 108 children with pre-diabetes (defined as the presence of multiple islet autoantibodies), the prevalence of vitamin D deficiency was higher in children with pre-diabetes than in controls. The cumulative incidence of T1DM at 10 years after seroconversion was similar between children with vitamin D deficiency and those with sufficient vitamin D levels. That is, vitamin D deficiency was not associated with faster progression to T1DM in children with multiple islet autoantibodies [68]. In addition, the Diabetes Autoimmunity Study in the Young (DAISY) was a prospective study conducted in Colorado (USA) involving 448 children between birth and 8 years of age. All were at increased risk of developing T1DM (HLA genotype or first-degree relatives of patients with T1DM). Autoantibodies and vitamin D status were checked regularly during follow-up. Their results showed that in children at increased risk of T1DM, there was no association between vitamin D status in infancy or throughout childhood and the risk of islet autoimmunity or progression to T1DM [69]. Finally, the Type 1 Diabetes Prevention and Prediction Study (DIPP) was a prospective cohort project in Finland with 252 children at increased risk of T1DM (HLA genotype and/or seroconversion to islet cell antibody positivity). During their follow-up (12 years), circulating vitamin D concentrations were measured from 3 months of age until diagnosis of T1DM. Their results showed that there was no difference in vitamin D status between children who progressed to T1DM and controls. In conclusion, that prospective study suggests that the development of T1DM is not associated with vitamin D status [70].

## 6. Vitamin D Supplementation in Type 1 Diabetes Mellitus

Experimental studies using non-obese diabetic mice as a model of human T1DM have demonstrated protective effects of vitamin D against islet autoimmunity and progressive β-cell destruction Also importantly, vitamin D deficiency in early life results in a higher incidence and earlier onset of diabetes. Calcitriol and its analogues have also been shown to prevent insulitis and thus diabetes, especially when given at an early age (when the immune attack on β-cells is in its early stages) [40,71,72]. In short, pre-clinical evidence suggest that vitamin D and its analogues appear to be able to protect β-cell mass and function from autoimmune attack by several mechanisms, including (i) promoting the shift from a Th1 to a Th2 cytokine expression profile and so decreasing the number of Th1 cells (autoreactive T cells), (ii) enhancing the clearance of autoreactive T cells and decreasing the Th1 cell infiltration within the pancreatic islets, (iii) reducing cytokine-induced β-cell damage, and (iv) promoting the differentiation and suppressive capacity of Treg cells.

The efficacy of vitamin D in halting or reversing islet autoimmunity observed in preclinical studies has stimulated numerous interventional studies and randomized controlled trials that have established beneficial clinical effects of different forms of vitamin D or analogues (in addition to insulin therapy) in patients with T1DM [40,51,73]. In other words, insulin therapy supplemented with vitamin D appears to be able improve the preservation of residual pancreatic β-cell function in patients with T1DM. Table 2 displays some relevant interventional studies conducted in children/adolescents with new-onset T1DM which resulted in significant preservation of residual pancreatic β-cell function and improved glycemic control.

Several interventional studies and randomized controlled trials have shown beneficial effects of cholecalciferol supplementation in addition to insulin therapy in children with new-onset T1DM, in terms of preservation of residual β-cell function and improvement of glycemic control [74,75,76]. In addition, cholecalciferol also appears to improve the suppressive function of Tregs, reduce autoantibody titers and daily insulin requirements [74,76]. However, cholecalciferol supplementation in children with established T1DM (disease duration > 12 months) did not show protective effects on pancreatic β-cell function or glycemic control [77,78,79]. In other words, the hypothesis that the protective effect of cholecalciferol supplementation is only visible when the disease duration is less than 1 year (patients with new-onset T1DM) seems to be confirmed [80].

Several studies on children with new-onset T1DM have reported potential protective effects on β-cell function from the use of cholecalciferol in combination with omega-3 polyunsaturated fatty acids (PUFAs), lansoprazole or sitagliptin. Omega-3 PUFAs (especially eicosapentaenoic acid (EPA) and docosahexaenoic acid (DHA)) have anti-inflammatory properties. In a placebo-controlled cohort study 26 children with new-onset T1DM, the combination of high-dose of cholecalciferol (1000 IU/day) and omega-3 PUFAs (EPA + DHA, 50–60 mg/kg/day) added to insulin therapy showed a reduction in insulin requirements after 12 months of supplementation. Therefore, these results suggest that this co-supplementation would preserve residual endogenous insulin secretion by attenuating autoimmunity and counteracting inflammation [81]. On the other hand, lansoprazole is a proton pump inhibitor (PPIs) that increase gastrin levels, and gastrin plays an important role in the regulation of β-cell neogenesis. Indeed, experimental and clinical studies have reported improved glycemic control in response to PPIs. In a placebo-controlled observational study of 14 children with new-onset T1DM, combination therapy with cholecalciferol (2000 IU/day) and lansoprazole 15 mg (<30 kg) or 30 mg (>30 kg) for six months in addition to insulin therapy was associated with a slower decline in residual β-cell function and lower insulin requirements [82]. Finally, sitagliptin is a dipeptidyl peptidase-4 inhibitor used as a hypoglycemic agent. It reduces α-cell glucagon secretion, increases β-cell insulin secretion and may stimulate β-cell proliferation. A recent retrospective case-control study in 46 children/adolescents with newly diagnosed T1DM showed that co-administration of cholecalciferol (5000 IU/day) plus sitagliptin (50 mg/day) in addition to insulin therapy can significantly prolong the duration of the clinical remission phase through synergistic anti-inflammatory and immunomodulatory properties [83]. Obviously, randomized controlled trials are needed to confirm whether these combinations can lead to preservation of β-cell function in children with new-onset T1DM.

There are few data on the efficacy of ergocalciferol in T1DM. However, a recent randomized, placebo-controlled trial showed that ergocalciferol supplementation (50,000 IU/week for 2 months, then fortnightly for 10 months) in addition to insulin therapy improved glycemic control and reduced serum TNF-α concentration and total daily insulin dose. In other words, these results suggest a protective action of ergocalciferol on residual β-cell function in children and adolescents with newly diagnosed T1DM [84]. However, larger studies are needed to quantify the effect of ergocalciferol in young people with T1DM.

There are also few data on the efficacy of calcidiol supplementation in newly diagnosed T1DM. A pilot intervention in 15 children with new-onset T1D who received calcidiol supplementation for one year (in addition to insulin treatment) showed a significant reduction in mononuclear reactivity against GAD-65 along with daily insulin dose, suggesting a potentially conservative effect of β-cells. Furthermore, calcidiol could modulate the immune response to β-cell autoantigens through local conversion to calcitriol (immune cells express 1α-hydroxylase and are therefore able to locally convert calcidiol to calcitriol) [85]. Unfortunately, the small sample size of these studies does not allow definitive conclusions to be drawn about the efficacy of calcidiol in T1DM; therefore, future prospective interventional studies should be conducted.

Studies of supplementation of the active form of vitamin D (calcitriol) in addition to insulin therapy for the treatment of new-onset T1DM have been disappointing. Two randomized placebo-controlled trials showed that calcitriol supplementation at a dose of 0.25 μg/day for 18 to 24 months was ineffective in preserving residual β-cell function and improving glycemic control in patients (children and young adults) with new-onset T1DM [86,87]. The negative results could be due to the doses of calcitriol administered in both studies, or the short serum half-life of calcitriol could lead to fluctuations in its serum concentration that would condition its immunomodulatory effects on β-cells. This means that larger studies with longer observation periods and calcitriol dose stratification are needed.

Alphacalcidol (1α-hydroxycholecalciferol) is a vitamin D analogue that is converted to calcitriol in the liver by the enzyme vitamin D-25-hydroxylase without the need for secondary renal hydroxylation. In addition, alphacalcidol improves immune senescence by acting as an anti-inflammatory agent through increased IL-10 and a decreased IL6/IL-10 ratio, and it also improves cellular immunity through an increased CD4/CD8 ratio [88]. Few studies have investigated the efficacy of alfacalcidol as an adjunct to insulin therapy in the treatment of T1DM. However, a randomized controlled trial confirmed that alfacalcidol (at 0.5 μg/day for six months) can safely preserve β-cell function in newly diagnosed T1DM in children [89].

**Table 2 ijms-26-04593-t002:** Interventional studies in children/adolescents with new-onset T1DM resulting in preservation of residual pancreatic β-cell function and improved glycemic control.

Author, Year, and Country	Study Design	Supplementation Dosage Duration	Significant Findings
Gabbay et al., 2012 (Brazil) [74]	Randomized, double blind, placebo-controlled, prospective trial	Cholecalciferol (2000 IU/d for 18 months)	Decrease in Hb1Ac levels Decrease in autoantibody titers Stimulated C-peptide enhancement Increase in Treg percentage
Ataie-Jafari et al., 2013 (Iran) [89]	Randomized, single blind, placebo-controlled trial	Alfacalcidol (0.25 μg/twice daily for 6 months)	Improved stimulated C-peptide
Federico et al., 2014 (Italy) [85]	Pilot interventional study	Calcidiol (10–30 μg/d for one year)	Decrease in insulin requirements Stability of fasting C-peptide levels Inhibition of GAD-65 antibodies
Treiber et al., 2015 (Austria) [75]	Randomized, double blind, placebo-controlled, prospective trial	Cholecalciferol (70 IU/kg/d for 12 months)	Decrease in Hb1Ac levels Stimulated C-peptide enhancement Reduction in daily insulin doses Increase in Treg percentage
Panjiyar et al., 2018 (India) [76]	Prospective, case-control, interventional study	Cholecalciferol (3000 IU/d for one year)	Decrease in Hb1Ac levels Reduction in daily insulin doses Stimulated C-peptide enhancement
Cadario et al., 2019 (Italy) [81]	Case study	Cholecalciferol (1000 IU/d) plus EPA + DHA (50–60 mg/kg/d) for 12 months	Decrease in insulin requirements
Reddy et al., 2022 (India) [82]	Pilot study	Cholecalciferol (2000 IU/d) plus lansoprazole (15–30 mg) for six months	Decrease in insulin requirements Slower fasting peptide-C decline
Pinheiro et al., 2023 (Brazil, Italy) [83]	Case-control study	Cholecalciferol (5000 IU/d) plus sitagliptin (50 mg/day) for 12 months	Longer duration of the remission phase
Nwosu et al., 2022 (Massachusetts, USA) [84]	Randomized, double blind, placebo-controlled, prospective trial	Ergocalciferol (50,000 IU/wk for 2 months, then fortnightly for 10 months)	Decrease in insulin requirements Reduction in TNF-α concentration

Unfortunately, in T1DM, the potential benefits of supplementation with vitamin D or analogues would be focused on preventing the onset of the disease rather than treating it, as the destruction of β-cells is irreversible. Indeed, as mentioned above, several randomized controlled trials in the last decade have shown that vitamin D supplementation, especially as cholecalciferol, appears to preserve residual β-cell function and improve glycemic control in children and adolescents with new-onset T1DM through its immunomodulatory and anti-inflammatory effects. However, future large-scale prospective trials are needed to adequately assess the role of vitamin D as an adjuvant treatment in T1DM.

To conclude, we would suggest a protocol for vitamin D supplementation in children and adolescents with new-onset T1DM based on the data in this narrative review. Obviously, serum calcidiol levels would be measured in all newly diagnosed T1DM patients. If calcidiol levels are less than 30 ng/mL, they should receive cholecalciferol supplementation (1000 to 2000 IU/day) to maintain a serum calcidiol concentration between 30 and 50 ng/mL (previously defined as optimal vitamin D levels). Patients with serum calcidiol concentrations above 30 ng/mL at the onset of diabetes should be monitored with serial calcidiol concentrations, and cholecalciferol supplementation should be initiated if serum calcidiol concentrations are below 30 ng/mL. Cholecalciferol supplementation should be continued for at least one year to ensure optimal vitamin D benefit. Of course, residual β-cell function (fasting plasma C-peptide levels), glycemic control (HbA1c levels and/or fasting plasma glucose), T1DM-associated autoantibodies (islet autoantibodies), and exogenous insulin requirements from diagnosis should be monitored periodically, together with vitamin D (calcidiol) levels.

On the other hand, research in recent years has suggested that vitamin D may also have an antioxidant effect by inhibiting the generation of free radicals, possibly by enhancing cellular glutathione levels [45,90]. However, the results of randomized controlled trials point to a controversial role of vitamin D as an antioxidant, as the potential role of vitamin D as an antioxidant could not be confirmed [91]. Additional research is required to clarify the possible effects of vitamin D supplementation on oxidative stress.

## 7. Conclusions

Interestingly, both the prevalence of T1DM and vitamin D deficiency are increasing worldwide and becoming serious public health problems. Accumulating data over the past decades suggest that vitamin D status is involved in the pathogenesis of T1DM. In fact, many researchers have found that vitamin D deficiency is more prevalent in children/adolescents with new-onset T1DM than in healthy individuals. Vitamin D is now known to have anti-inflammatory and immunomodulatory effects based on the control of gene transcription, which is crucial for the maintenance of self-tolerance. Therefore, vitamin D deficiency should be considered as an important environmental factor for the development of T1DM.

On the other hand, results from long-term follow-up studies in children suggest that there is no association between vitamin D status before diagnosis and the occurrence of T1DM later in life. In addition, the potential role of vitamin D as an adjuvant therapy in T1DM remains inconclusive. Indeed, the variability of results in human trials is generally frustrating, and, at best, it has a short-term positive effect on newly diagnosed T1DM patients. Most of the researchers conclude their papers by pointing out that large-scale prospective randomized controlled trials are urgently needed to definitively establish the role of vitamin D in T1DM. Discrepancies in clinical outcomes have been attributed to a variety of reasons, including heterogeneity in the design or the population studied, as well as different formulations and doses of vitamin D used, the duration of different trials, and, in many cases, small sample sizes in the studies.

We believe that the fact that interventional studies generally have better esults in patients newly diagnosed with diabetes should not be surprising. In our opinion, the weakness of the aforementioned interventional studies does not lie in the heterogeneity of their characteristics, but rather in their relationship with functional or residual β-cell mass To date, all intervention studies have been carried out in the clinical phase of T1DM, whereas it would be desirable to be able to do so in the early stages of the autoimmune process (pre-diabetes). In other words, the efforts of the desired randomized controlled trials should also be directed towards obtaining biomarkers that can detect the onset of autoimmune insulitis and, in these precise circumstances, initiate vitamin D or analogue supplementation in addition to insulin treatment. This could possibly slow down or prevent the autoimmune process from continuing its progressive course and, as far as possible, avoid reaching the clinical and irreversible phase of the disease.

## Figures and Tables

**Figure 1 ijms-26-04593-f001:**
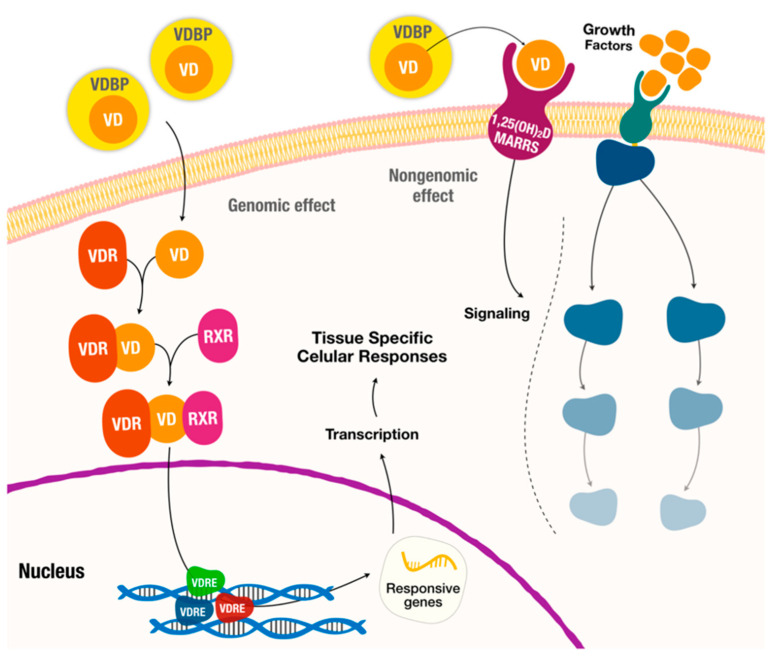
Hormonal actions of vitamin D with genomic and nongenomic effects (adapted from [5]). VDBP: vitamin D binding protein. VD: vitamin D. MARRS: membrane-associated rapid response steroid-binding protein. VDR: vitamin D receptor. RXR: retinoid X receptor. VDRE: vitamin D response elements.

**Figure 2 ijms-26-04593-f002:**
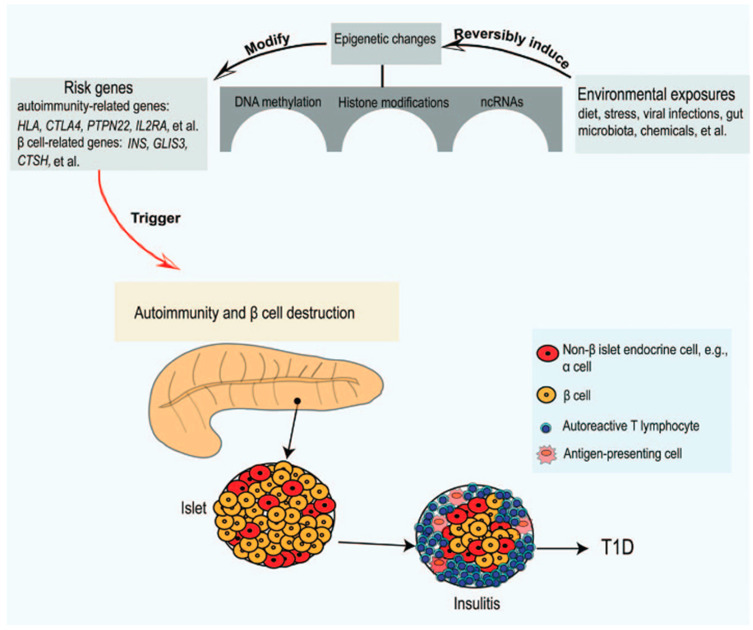
Environmental exposures induce epigenetic changes in genetically predisposed subjects to trigger the autoimmune destruction of the pancreatic β-cells (adapted from [28]).

**Table 1 ijms-26-04593-t001:** Immunomodulatory effects of vitamin D on immune cells.

Immune Cell Type	Vitamin-D-Induced Effect
Macrophages	▼ Pro-inflammatory IL-1, IL-6, IL-8, IL-12 ▲ Anti-inflammatory IL-10
Natural killer cells	▼ Pro-inflammatory IFN-γ ▲ Anti-inflammatory IL-4
Dendritic cells	▼ Pro-inflammatory IL-2, IL-6, IL-12 ▲ Anti-inflammatory IL-10 ▼ DCs’ differentiation (tolerogenic DCs) ▼ Antigen-presenting cells (T-cell anergy)
CD4+ T cells	▼ Hyperactivation ▼ Th1, Th17 ▲ Th2, Treg ▲ Anti-inflammatory IL-4, IL-5, IL-10, TGF-β ▼ Pro-inflammatory IL-2, IFN-γ, TNF-α, IL-17, IL-21
CD8+ T cells	▼ Hyperactivation
B cells	▼ B cell proliferation and differentiation into plasma cells ▼ Memory B cell formation ▼ Autoantibody production

Ref.: Infante et al., 2019 [40]. He et al., 2022 [41]. Prietl et al., 2013 [49]. Dankers et al., 2017 [50]. Rak et al., 2018 [51]. Cyprian et al., 2019 [52]. Gallo et al., 2023 [53]. Ghaseminejad-Raeini et al., 2023 [54]. Park et al., 2024 [5]. Galdo-Torres et al., 2025 [55].

## Data Availability

The datasets generated during and/or analyzed during the current study are available from the corresponding author upon reasonable request.

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
