# Peer review of "Type 1 Diabetes Mellitus and Vitamin D"

_ijms, 2025, doi:10.3390/ijms26104593_

Round 1

Reviewer 1 Report

Comments and Suggestions for Authors

The manuscript entitled: Type 1 Diabetes Mellitus and Vitamin D, is interesting and timely, which could help to analyze the current state of knowledge between both variables. However, this manuscript contains abundant repetitive ideas, contradictory premises that are not analyzed, mixture of preclinical and clinical evidence, disorder in the ideas, which may confuse the reader and generate more uncertainty.

I suggest correcting the order of the ideas and deepening the analysis of the contrasting evidence. Also, to recognize the methodological weaknesses and possible biases inherent to a narrative review.

Specifically, in each section it is necessary to answer or correct what follows.

Introduction

A) In the first paragraph, the relationship between vitamin D deficiency and the etiopathogenesis of diabetes mellitus is established as a real fact, and it is suggested to explore the knowledge gap in the contrasting studies that make this research necessary.

B) In the second paragraph, it is suggested to deepen in a more general way on the non-genomic mechanisms that vitamin D exerts on cytokine synthesis.

C) The third paragraph, the information on the oxidative stress that characterizes diabetes 1 and the antioxidant properties of vitamin d should be expanded, with the aim of describing a greater number of elements that support the knowledge gap to justify the present review.

D) To consult past reviews, such as those of 2005, with the aim of highlighting the novel contributions and the main advances that have been made over the last 20 years.

Mathieu C, Badenhoop K. Vitamin D and type 1 diabetes mellitus: state of the art. Trends Endocrinol Metab. 2005 Aug;16(6):261-6. doi: 10.1016/j.tem.2005.06.004. PMID: 15996876.

E) The objective is somewhat ambiguous, especially in the first part (environmental risk factory) and discordant with the objective that appears in the summary.

Vitamin D Synthesis and Metabolism

a) In the absorption of vitamin D, it is suggested to describe the importance of intestinal microbiota.

b) It is necessary to enrich the non-genomic mechanisms that vitamin D exerts specifically on the antioxidant effect or on the production of nitric oxide.

c) In the genomic mechanisms, focus on illustrating the specific mechanism on cytokine production (Figure 1).

d) To deepen in the explanation of the heterodimerization phenomenon and its impact on the functionality of nuclear receptors. As described, it seems that this phenomenon is simple, however, it is more complex than one imagines. Please expand on the heterodimerization phenomenon.

Pathogenesis and Natural History of Type 1 Diabetes Mellitus

A) Avoid using repetitive premises such as the one described in the first three lines of the abstract and in lines 106-107 and 117-118.

B) Subsequent to the following paragraph it is necessary to raise, what could represent the development of molecular experiments based on cell culture and viruses related to the pathophysiology of type 1 diabetes.

However, it is extremely difficult to prove a cause-effect relationship be-141 tween viral infections and the development of T1DM, as the period between viral expo-142 sure and the onset of clinical symptoms of T1DM is often very long.

C) I think there is a disorder in how the pathogenesis of diabetes is approached:

It is suggested first to clarify that the phenomenon of autoimmunity is related both to factors proper to the organism and to environmental factors and the interaction of both.

In that order, first address the factors of the organism itself such as genetic predispositions and then continue with the environmental factors: toxins and viruses.

Next, address the interactions of both variables: nutritional factors, including (breastfeeding, early food consumption in infancy, vitamin d intake). In addition, the role of the microbiota, that may be influenced by diet and how dysbiosis, avitaminosis correlate with autoimmunity, until the end addressing epigenetics.

D) In line 171, only the phenomenon of immunological tolerance is mentioned without defining the concept, it is necessary to recognize that this phenomenon is complex and is integrated by several variables.

E) In line 197 there is a grammatical error.

F) The information in lines 209-210 is repetitive with that previously shown in line 143.

Immunomodulatory Effects of Vitamin D in Autoimmune Diseases

a) To deepen the mechanism of action of vitamin D on cytokine production.

b) The information in lines 227-228 is somewhat repetitive with lines 236 -237.

Vitamin D Status and the Risk of T1D

A) To better contrast this analysis, I suggest integrating another study describing lack of differences in serum vitamin D3 levels.

Polat İ, Can Yılmaz G, Dedeoğlu Ö. Vitamin D and Nerve Conduction In Pediatric Type-1 Diabetes Mellitus. Brain Dev. 2022 May;44(5):336-342. doi: 10.1016/j.braindev.2022.01.001. Epub 2022 Jan 15. PMID: 35042650.

B) In relation to the information in the following premise:

Specifically, vitamin D deficiency appears to be much more prevalent in 267 patients with T1DM than in healthy individuals [13].

It is suggested to analyze and integrate the information contained in the following study:

Hou Y, Song A, Jin Y, Xia Q, Song G, Xing X. A dose-response meta-analysis between serum concentration of 25-hydroxy vitamin D and risk of type 1 diabetes mellitus. Eur J Clin Nutr. 2021 Jul;75(7):1010-1023. doi: 10.1038/s41430-020-00813-1. Epub 2020 Nov 24. PMID: 33235321; PMCID: PMC8266682.

C) In the premises described between lines 272- 274 and 276-278 there are no bibliographic citations.

D) In paragraph 287 to 289, two studies are described and the reference of only one study is cited.

E) In the following paragraph, there are no bibliographic references for either of the two premises. Avoid combining evidence from animal and human studies:

Experimental studies using non-obese diabetic mice as a model of human T1DM have demonstrated protective effects of vitamin D against islet autoimmunity and progressive β-cell dysfunction. Also importantly, vitamin D deficiency in early life results in a higher incidence and earlier onset of diabetes. Calcitriol and its analogues have also been shown 371 to prevent insulitis and thus diabetes, especially when given at an early age (when the 372 immune attack on β-cells is in its early stages).

F) In table 1 it is suggested to integrate one more column with a summary of the main results of each study.

G) The following paragraphs are somewhat repetitive and with contrasting ideas:

In addition, alphacalcidol improves immune senescence 455 by acting as an anti-inflammatory agent through increased IL-10 and decreased IL6/IL-10 456 ratio, and also improves cellular immunity through increased CD4/CD8 ratio [81]. Few

Indeed, as mentioned above, several 464 randomised controlled trials in the last decade have shown that vitamin D supplementa-465 tion, especially as cholecalciferol, appears to preserve residual β-cell function and im-466 prove glycaemic control in children and adolescents with new-onset T1DM through its

H) It is also suggested to consider the prevalence of patients with type 1 diabetes in underdeveloped countries. Where it would be difficult to measure vitamin D levels. Based on the above, what options or suggestions would you have for treating physicians in these countries?

I) For the above, also, the main weaknesses of a narrative review should be recognized; alluding to the possible biases involved in the classification of the studies considered, trying to have solid arguments to minimize the influence of these bias factors.

J) The last conclusion is inappropriate, since it is a suggestion and does not agree with the main objective.

K) The algorithm in the form of a diagram is inappropriate, if the criteria are not standardized in a larger number of studies and the unfeasibility of these criteria in underdeveloped countries is not recognized.

Comments on the Quality of English Language

OK

Author Response

Responses to reviewer-1

First of all, we would like to thank you for your suggestions as well as your words of encouragement regarding this article.

First of all, we would like to thank you for your advice regarding this article

NOTE: The corrected text of the new version is in red

Comments and Suggestions for Authors

The manuscript entitled: Type 1 Diabetes Mellitus and Vitamin D, is interesting and timely, which could help to analyze the current state of knowledge between both variables. However, this manuscript contains abundant repetitive ideas, contradictory premises that are not analyzed, mixture of preclinical and clinical evidence, disorder in the ideas, which may confuse the reader and generate more uncertainty.

I suggest correcting the order of the ideas and deepening the analysis of the contrasting evidence. Also, to recognize the methodological weaknesses and possible biases inherent to a narrative review.

Specifically, in each section it is necessary to answer or correct what follows.

Introduction

A) In the first paragraph, the relationship between vitamin D deficiency and the etiopathogenesis of diabetes mellitus is established as a real fact, and it is suggested to explore the knowledge gap in the contrasting studies that make this research necessary.

The Introduction sets out the background to the study and defines the objectives. We have tried to follow a logical sequence with bibliographic support.

The first paragraph does not make any definitive or exhaustive statements. For example:

Lines 37-38: “...leading to the hypothesis that vitamin D may play a role in the T1DM process.” [Chen et al., 2017. Park et l., 2024]

Lines: 40-41: “This also suggests that people living at high latitudes may be predisposed to T1DM.;...” [Hou et al., 2021]

Lines: 43-44: “...observational studies have shown that children with new-onset T1DM tend to have significantly lower vitamin D levels than healthy controls children.” [Franchi et al., 2014, Feng et al., 2015. Al-Zubeidi et al., 2016. Rasoul et al., 2016, Marino et al., 2019. Botelho et al, 2020, Daskalopoulou et al., 2022. Wu et al.,, 2023. Jacobs et al., 2024. Yang et al., 2024, Park et al., 2024,]

B) In the second paragraph, it is suggested to deepen in a more general way on the non-genomic mechanisms that vitamin D exerts on cytokine synthesis.

The present review is aimed at clinicians. We consider that it is beyond the scope of this review to delve into the non-genomic mechanisms that vitamin D exerts on cytokine synthesis.

C) The third paragraph, the information on the oxidative stress that characterizes diabetes 1 and the antioxidant properties of vitamin d should be expanded, with the aim of describing a greater number of elements that support the knowledge gap to justify the present review.

Vitamin D is considered a pleiotropic hormone and, of course, includes an antioxidant role. However, this role does not appear to be involved in the etiopathonesis of T1DM.

The etiopathogenesis of type 1 and type 2 diabetes are different.

T2DM (approximately 90% of all cases of diabetes) is a chronic metabolic disorder characterized by insulin resistance and/or and beta-cell dysfunction and consequentially high blood glucose. Under hyperglycaemic conditions, reactive oxygen species (hydrogen peroxide and superoxide anion) are likely to be involved in pancreatic beta-cell dysfunction and insulin resistance. Hyperglycaemia-induced oxidative stress would play a substantial role in the complications of this disease. Indeed, there is increasing awareness and evidence that diabetes mellitus, particularly type 2 diabetes, is significantly modulated by oxidative stress.

There is a recent narrative review that is very illustrative on this issue: Singh, et al. Mechanistic Insight into Oxidative Stress-Triggered Signaling Pathways and Type 2 Diabetes. Molecules 2022, 27, 950.

On the other hand, T1DM -subject of this review- is induced by destruction of pancreatic beta-cells which is mediated by an autoimmune mechanism and consequent inflammatory process. Evidently, poorly controlled T1DM will cause oxidative stress and likely contribute to the development of subsequent vascular complications. However, due to the low functional islet mass at the onset of the disease, it does not appear to substantially contribute to the functional impairment of pancreatic beta cells.

D) To consult past reviews, such as those of 2005, with the aim of highlighting the novel contributions and the main advances that have been made over the last 20 years.

Mathieu C, Badenhoop K. Vitamin D and type 1 diabetes mellitus: state of the art. Trends Endocrinol Metab. 2005 Aug;16(6):261-6. doi: 10.1016/j.tem.2005.06.004. PMID: 15996876.

This reference is excellent and very clear-sighted, but in the text we have limited ourselves to more current references.

E) The objective is somewhat ambiguous, especially in the first part (environmental risk factory) and discordant with the objective that appears in the summary.

As described in the text (Pathogenesis and Natural History of T1DM), T1DM is a complex multifactorial disease in which environmental factors and genetic predisposition interact to promote the induction of an autoimmune response against β-cells. Viral infections have been considered the most important environmental factors in initiating the process of autoimmune destruction of β-cells.Viruses can damage pancreatic β-cells by inducing an autoimmune response against β-cells (phenomenon of molecular mimicry). In recent years, in addition to viral infections, several epidemiological studies have described new environmental factors, including vitamin D deficiency, which may play a role in triggering islet autoimmunity and T1DM in subjects at high genetic risk. That is, vitamin D deficiency may play a role in the pathogenesis of T1DM

Therefore, we understand that there is no discordance between the paragraphs of the abstract: “...about the association between vitamin D status in the pathogenesis of T1DM.” and the Introduction: “...(a) research progress on the possible function of vitamin D status as an environmental risk factor in the pathogenesis of T1DM.”

Vitamin D Synthesis and Metabolism

a) In the absorption of vitamin D, it is suggested to describe the importance of intestinal microbiota.

Although the importance of the intestinal microbiota in the absorption of vitamin D is a very interesting topic, we understand that it is not the preferred subject of this review.

b) It is necessary to enrich the non-genomic mechanisms that vitamin D exerts specifically on the antioxidant effect or on the production of nitric oxide.

As previously discussed, the antioxidant effect of vitamin D does not appear to be involved in the etiopathogenesis of T1DM.

c) In the genomic mechanisms, focus on illustrating the specific mechanism on cytokine production (Figure 1).

As mentioned above, this review is oriented to clinicians. Therefore, it is beyond the scope of this review to delve into the non-genomic mechanisms that vitamin D exerts on cytokine synthesis.

d) To deepen in the explanation of the heterodimerization phenomenon and its impact on the functionality of nuclear receptors. As described, it seems that this phenomenon is simple, however, it is more complex than one imagines. Please expand on the heterodimerization phenomenon.

Of course, this is a complex phenomenon, but we believe that it would be beyond the scope of this review to go into its molecular explanation.

Pathogenesis and Natural History of Type 1 Diabetes Mellitus

A) Avoid using repetitive premises such as the one described in the first three lines of the abstract and in lines 106-107 and 117-118.

It is inevitable that there may be some repetition between the Abstract and some sections of the article.

B) Subsequent to the following paragraph it is necessary to raise, what could represent the development of molecular experiments based on cell culture and viruses related to the pathophysiology of type 1 diabetes.

However, it is extremely difficult to prove a cause-effect relationship between viral infections and the development of T1DM, as the period between viral exposure and the onset of clinical symptoms of T1DM is often very long.

Unfortunately, our experience is exclusively clinical and we are not able to design molecular experiments in this field.

C) I think there is a disorder in how the pathogenesis of diabetes is approached:

It is suggested first to clarify that the phenomenon of autoimmunity is related both to factors proper to the organism and to environmental factors and the interaction of both.

In that order, first address the factors of the organism itself such as genetic predispositions and then continue with the environmental factors: toxins and viruses.

Next, address the interactions of both variables: nutritional factors, including (breastfeeding, early food consumption in infancy, vitamin d intake). In addition, the role of the microbiota, that may be influenced by diet and how dysbiosis, avitaminosis correlate with autoimmunity, until the end addressing epigenetics.

Part of the text has been deleted. From “Vitamin D deficiency is common in the pediatric population with T1DM....” to “...vitamin D in early life reduces the risk of diabetes” (lines 151-162 of the first version).

And now the sequence of section 3 (Pathogenesis and natural history of T1DMD) is quite similar to that advised by the reviewer:

(1) Reference is made to: “the genetic predisposition to β-cell autoimmunity in individuals with specific human leukocyte antigen (HLA-DR3-DQ2 or HLA-DR4-DQ8), and other genes related to autoimmunity (CTLA4, PTPN22, L2RA, etc.) or to β-cells (INS, GLIS3, CTSH, etc.).”

(2) Then it is said: “...the selective destruction of β-cells would be caused by an interaction between risk genes and environmental factors.” To date, viral infections have been considered the most important environmental factors in initiating the process of autoimmune destruction of β-cells (phenomenon of molecular mimicry).”

(3) And it continues: “In recent years, in addition to viral infections, several epidemiological studies have described new environmental factors (cow's milk, gluten or Omega-3 fatty acids intake, dysbiosis, vitamin D deficiency, etc.) which may play a role in triggering islet autoimmunity and T1DM in subjects at high genetic risk.”

(4) To finish with: “All of these factors could alter gene expression through epigenetic mechanisms (particularly DNA methylation, histone modifications or long non-coding RNA) thereby inducing an aberrant immune response and progressive β-cells destruction (Figure 2), and thus be involved in the pathogenesis of T1DM.”

D) In line 171, only the phenomenon of immunological tolerance is mentioned without defining the concept, it is necessary to recognize that this phenomenon is complex and is integrated by several variables.

The concept of immunological tolerance is commonly used, and we have not been forced to define it.

In short: specific absence of response of the immune system to a self-antigen. This is an active state (not a simple absence of response), endowed with specificity and memory.

E) In line 197 there is a grammatical error.

Previous text... A higher proportion of plasma cells into insulitis has been associated with a faster β-cell decline [39].”

has been changed to...”A higher proportion of plasma cells in insulitis has been associated with a faster β-cell decline [39].”

F) The information in lines 209-210 is repetitive with that previously shown in line 143.

We consider that the sentences mentioned are not redundant and that both fit in the text.

Line 146-147: “as the period between viral exposure and the onset of clinical symptoms of T1DM is often very long.”

Lines: 205-208): “ usually appear several years after the onset of the autoimmune process, when most of the pancreatic β-cells have been destroyed (stage 3), and T1DM is definitively established”

Immunomodulatory Effects of Vitamin D in Autoimmune Diseases

a) To deepen the mechanism of action of vitamin D on cytokine production.

It is beyond the scope of this article to delve into the mechanism of vitamin D on cytokine production. In this section we have tried to simplify the immunomodulatory effects of vitamin D on various immune cell lineages. In fact, a brief summary (table 1) on the immunomodulatory effects of vitamin D has been added for better understanding of the text.modulatory effects of vitamin D has been added for better understanding of the text

b) The information in lines 227-228 is somewhat repetitive with lines 236 -237.

The information on lines 233-239 has been modified.

Previous text... “Vitamin D promotes the differentiation of T-helper lymphocytes from a Th1 and Th17 profile to a Th2 and Treg profile, increasing the release of anti-inflammatory cytokines such as IL-4, IL-5 and IL10, while decreasing the production of pro-inflammatory cytokines, including IL-2, IFN-γ, TNF-α, IL-17 and IL-21.”

Has been changed to...“Vitamin D promotes the differentiation of T-helper lymphocytes from a Th1 to a Th2 phenotype. This change implies inhibition of inflammatory cytokine production (IL-2, IFN-γ, and TNF-α) and an increased production of anti-inflammatory cytokines (IL-4, IL-5, and IL-10). Furthermore, vitamin D affects differentiation to Th17 phenotype, leading to a decrease in the production of inflammatory cytokines (IL-17 and IL-21), and facilitates the induction of T reg cells with increased production of anti-inflammatory cytokines (IL-10 and TGF-β).”

Vitamin D Status and the Risk of T1D

A) To better contrast this analysis, I suggest integrating another study describing lack of differences in serum vitamin D3 levels.

Polat İ, Can Yılmaz G, Dedeoğlu Ö. Vitamin D and Nerve Conduction In Pediatric Type-1 Diabetes Mellitus. Brain Dev. 2022 May;44(5):336-342. doi: 10.1016/j.braindev.2022.01.001. Epub 2022 Jan 15. PMID: 35042650.

It seems to us a very interesting reference (their results suggest that hypovitaminosis D could lead to the development of neuropathic changes already in the early stages of the disease) that we will take into account in the evolutionary control of our patients with T1DM.

However, we believe that this specific topic is beyond the scope of this review.

B) In relation to the information in the following premise:

Specifically, vitamin D deficiency appears to be much more prevalent in patients with T1DM than in healthy individuals [13].

It is suggested to analyze and integrate the information contained in the following study:

Hou Y, Song A, Jin Y, Xia Q, Song G, Xing X. A dose-response meta-analysis between serum concentration of 25-hydroxy vitamin D and risk of type 1 diabetes mellitus. Eur J Clin Nutr. 2021 Jul;75(7):1010-1023. doi: 10.1038/s41430-020-00813-1. Epub 2020 Nov 24. PMID: 33235321; PMCID: PMC8266682.

Sorry, this sentence has been deleted due to an inadvertent typographical error.

C) In the premises described between lines 272- 274 and 276-278 there are no bibliographic citations.

References cited subsequently (Infante et al., 2019. Manousaki et al., 2021. Najjar et al., 2021. Wang et al., 2014. Habibian et al., 2019. Ran et al., 2021) suggest that vitamin D status may be an important environmental risk factor in the pathogenesis of T1DM, rather than a consequence of pathophysiological changes resulting from the disease.

D) In paragraph 287 to 289, two studies are described and the reference of only one study is cited.

This sentence only referred to a study: [51] Najjar et al. Vitamin D and Type 1 Diabetes Risk: A. Systematic Review and Meta-Analysis of Genetic Evidence. Nutrients 2021, 13, 4260.

E) In the following paragraph, there are no bibliographic references for either of the two premises. Avoid combining evidence from animal and human studies:

Experimental studies using non-obese diabetic mice as a model of human T1DM have demonstrated protective effects of vitamin D against islet autoimmunity and progressive β-cell dysfunction. Also importantly, vitamin D deficiency in early life results in a higher incidence and earlier onset of diabetes. Calcitriol and its analogues have also been shown to prevent insulitis and thus diabetes, especially when given at an early age (when the immune attack on β-cells is in its early stages).

We had placed the bibliographical references in the wrong place. In addition, a new reference has been added. Previous text has been changed to:

Experimental studies using non-obese diabetic mice as a model of human T1DM have demonstrated protective effects of vitamin D against islet autoimmunity and progressive β-cell dysfunction. Also importantly, vitamin D deficiency in early life results in a higher incidence and earlier onset of diabetes. Calcitriol and its analogues have also been shown to prevent insulitis and thus diabetes, especially when given at an early age (when the immune attack on β-cells is in its early stages) [38, 64, 65].”

We have avoided combining evidence from animal and human studies. In fact, this paragraph serves as a preamble to the next paragraph:

The efficacy of vitamin D in halting or reversing islet autoimmunity observed in preclinical studies has stimulated numerous interventional studies and randomised controlled trials that have established beneficial clinical effects of different forms of vitamin D or analogues (in addition to insulin therapy) in patients with T1DM...”

F) In table 1 it is suggested to integrate one more column with a summary of the main results of each study.

A new column has been added to Table 2 with the most significant findings in each of the interventional studies.

Table 1. Interventional studies in children/adolescents with new-onset T1DM resulting in preservation of residual pancreatic β-cell function and improved glycaemic control.

Author, Year and Country

Study design

Suplementation dosage Duration

Significants findings

Gabbay et al., 2012 (Brazil) [67]

Randomized, double blind, placebo-controlled, prospective trial

Cholecalciferol (2000 IU/d for 18 months)

Decrease in Hb1Ac levels

Decrease in autoantibody titers

Stimulated C-peptide enhancement

Increase Treg percentage

Ataie-Jafari et al., 2013 (Iran) [82]

Randomized, single blind, placebo-controlled trial

Alfacalcidol (0.25 ug/twice daiy for 6 months)

Improved stimulated C-peptide

Federico et al, 2014 (Italy) [78]

Pilot interventional study

Calcidiol (10-30 ug/d for one year)

Decrease insulin requirements

Stability of fasting C-peptide levels

Inhibition of GAD-65 antibodies

Treiber et al., 2015 (Austria) [68]

Randomized, double blind, placebo-controlled, prospective trial

Cholecalciferol (70 IU/kg/d for 12 months)

Decrease in Hb1Ac levels

Stimulated C-peptide enhancement

Reduction in daily insulin doses

Increase Treg percentage

Panjiyar et al., 2018 (India) [69]

Prospective, case-control, interventional study

Cholecalciferol (3000 IU/d for one year)

Decrease in Hb1Ac levels

Reduction in daily insulin doses

Stimulated C-peptide enhancement

Cadario et al., 2019 (Italy) [74]

Cases study

Cholecalciferol (1000 IU/d) plus EPA+DHA (50-60 mg/kg/d) for 12 months

Decrease insulin requirements

Reddy et al., 2022 (India) [75]

Pilot study

Cholecalciferol (2000 IU/d) plus lansoprazole (15-30 mg) for six months)

Decrease insulin requirements

Slower fasting peptide-C decline

Pinheiro et al., 2023 (Brasil, Italy) [76]

Case-control study

Cholecalciferol (5,000 IU/d) plus sitagliptin (50 mg/day) for 12 months

Longer duration of the remission phase

Nwosu et al., 2024 (Massachusetss, USA) [77]

Randomized, double blind, placebo-controlled, prospective trial

Ergocalciferol (50,000 IU/wk for 2 months, then fortnightly for 10 months)

Decrease insulin requirements

Reduction TNF-α concentration

G) The following paragraphs are somewhat repetitive and with contrasting ideas:

In addition, alphacalcidol improves immune senescence by acting as an anti-inflammatory agent through increased IL-10 and decreased IL6/IL-10 ratio, and also improves cellular immunity through increased CD4/CD8 ratio [81]. Few

Indeed, as mentioned above, several randomised controlled trials in the last decade have shown that vitamin D supplementa tion, especially as cholecalciferol, appears to preserve residual β-cell function and im prove glycaemic control in children and adolescents with new-onset T1DM through its

We are sorry, but we do not see any conflicting ideas between these two texts. The second text can be considered as the epilogue to section 6 (Vitamin D Supplementation in Type 1 Diabetes).

H) It is also suggested to consider the prevalence of patients with type 1 diabetes in. Where it would be difficult to measure vitamin D levels. Based on the above, what options or suggestions would you have for treating physicians in these countries?

In the bibliographic review we have found published data on this subject carried out in countries of the five continents.

As noted in section 7 (Conclusions): “Most of the researchers conclude their papers by pointing out that large-scale prospective randomized controlled trials are urgently needed to definitively establish the role of vitamin D in T1DM.” When this happens we will be in a position to suggest qualified options to clinicians in any country.

I) For the above, also, the main weaknesses of a narrative review should be recognized; alluding to the possible biases involved in the classification of the studies considered, trying to have solid arguments to minimize the influence of these bias factors.

Narrative reviews are ideal for presenting general aspects of a given topic, but conceptually they carry a great weakness (subjectivity and/or unclear methodology).

To reduce bias, many more articles have been analyzed than those listed as references.

J) The last conclusion is inappropriate, since it is a suggestion and does not agree with the main objective.

Surprisingly, in the bibliographic review carried out, no author has hinted at this possibility which, on the other hand, we feel obliged to point out. Moreover, we are proud to be able to do so on the basis of an extensive bibliographic analysis:

We believe that the fact that interventional studies generally have better results in patients newly diagnosed with diabetes should not be surprising. This should lead us to believe that the weakness of the aforementioned interventional studies does not lie in the heterogeneity of their characteristics, but rather in their relationship with functional or residual β-cell mass. To date, all intervention studies have been carried out in the clinical phase of T1DM, whereas it would be desirable to be able to do so in the early stages of the autoimmune process (pre-diabetes). In other words, the efforts of the desired randomised controlled trials should also be directed towards obtaining biomarkers that can detect the onset of autoimmune insulitis and, in these precise circumstances, initiate vitamin D or analogues supplementation in addition to insulin treatment. This could possibly slow down or prevent the autoimmune process from continuing its progressive course and, as far as possible, avoid reaching the clinical and irreversible phase of the disease.”

K) The algorithm in the form of a diagram is inappropriate, if the criteria are not standardized in a larger number of studies and the unfeasibility of these criteria in underdeveloped countries is not recognized.

Completely in agreement. We have deleted the algorithm, but kept the explanatory text. Most of the researchers conclude their papers by pointing out that large-scale prospective randomized controlled trials are urgently needed to definitively establish the role of vitamin D in T1DM. In the meantime, we would suggest a protocol for vitamin D supplementation in children and adolescents with new-onset T1DM based on the data in this narrative review

We would like to express our thanks to referee for your suggestions and positive criticisms.

We hope every made question have been answered adequately.

Yours sincerely,

Responses to reviewer-1

First of all, we would like to thank you for your suggestions as well as your words of encouragement regarding this article.

First of all, we would like to thank you for your advice regarding this article

NOTE: The corrected text of the new version is in red

Comments and Suggestions for Authors

The manuscript entitled: Type 1 Diabetes Mellitus and Vitamin D, is interesting and timely, which could help to analyze the current state of knowledge between both variables. However, this manuscript contains abundant repetitive ideas, contradictory premises that are not analyzed, mixture of preclinical and clinical evidence, disorder in the ideas, which may confuse the reader and generate more uncertainty.

I suggest correcting the order of the ideas and deepening the analysis of the contrasting evidence. Also, to recognize the methodological weaknesses and possible biases inherent to a narrative review.

Specifically, in each section it is necessary to answer or correct what follows.

Introduction

A) In the first paragraph, the relationship between vitamin D deficiency and the etiopathogenesis of diabetes mellitus is established as a real fact, and it is suggested to explore the knowledge gap in the contrasting studies that make this research necessary.

The Introduction sets out the background to the study and defines the objectives. We have tried to follow a logical sequence with bibliographic support.

The first paragraph does not make any definitive or exhaustive statements. For example:

Lines 37-38: “...leading to the hypothesis that vitamin D may play a role in the T1DM process.” [Chen et al., 2017. Park et l., 2024]

Lines: 40-41: “This also suggests that people living at high latitudes may be predisposed to T1DM.;...” [Hou et al., 2021]

Lines: 43-44: “...observational studies have shown that children with new-onset T1DM tend to have significantly lower vitamin D levels than healthy controls children.” [Franchi et al., 2014, Feng et al., 2015. Al-Zubeidi et al., 2016. Rasoul et al., 2016, Marino et al., 2019. Botelho et al, 2020, Daskalopoulou et al., 2022. Wu et al.,, 2023. Jacobs et al., 2024. Yang et al., 2024, Park et al., 2024,]

B) In the second paragraph, it is suggested to deepen in a more general way on the non-genomic mechanisms that vitamin D exerts on cytokine synthesis.

The present review is aimed at clinicians. We consider that it is beyond the scope of this review to delve into the non-genomic mechanisms that vitamin D exerts on cytokine synthesis.

C) The third paragraph, the information on the oxidative stress that characterizes diabetes 1 and the antioxidant properties of vitamin d should be expanded, with the aim of describing a greater number of elements that support the knowledge gap to justify the present review.

Vitamin D is considered a pleiotropic hormone and, of course, includes an antioxidant role. However, this role does not appear to be involved in the etiopathonesis of T1DM.

The etiopathogenesis of type 1 and type 2 diabetes are different.

T2DM (approximately 90% of all cases of diabetes) is a chronic metabolic disorder characterized by insulin resistance and/or and beta-cell dysfunction and consequentially high blood glucose. Under hyperglycaemic conditions, reactive oxygen species (hydrogen peroxide and superoxide anion) are likely to be involved in pancreatic beta-cell dysfunction and insulin resistance. Hyperglycaemia-induced oxidative stress would play a substantial role in the complications of this disease. Indeed, there is increasing awareness and evidence that diabetes mellitus, particularly type 2 diabetes, is significantly modulated by oxidative stress.

There is a recent narrative review that is very illustrative on this issue: Singh, et al. Mechanistic Insight into Oxidative Stress-Triggered Signaling Pathways and Type 2 Diabetes. Molecules 2022, 27, 950.

On the other hand, T1DM -subject of this review- is induced by destruction of pancreatic beta-cells which is mediated by an autoimmune mechanism and consequent inflammatory process. Evidently, poorly controlled T1DM will cause oxidative stress and likely contribute to the development of subsequent vascular complications. However, due to the low functional islet mass at the onset of the disease, it does not appear to substantially contribute to the functional impairment of pancreatic beta cells.

D) To consult past reviews, such as those of 2005, with the aim of highlighting the novel contributions and the main advances that have been made over the last 20 years.

Mathieu C, Badenhoop K. Vitamin D and type 1 diabetes mellitus: state of the art. Trends Endocrinol Metab. 2005 Aug;16(6):261-6. doi: 10.1016/j.tem.2005.06.004. PMID: 15996876.

This reference is excellent and very clear-sighted, but in the text we have limited ourselves to more current references.

E) The objective is somewhat ambiguous, especially in the first part (environmental risk factory) and discordant with the objective that appears in the summary.

As described in the text (Pathogenesis and Natural History of T1DM), T1DM is a complex multifactorial disease in which environmental factors and genetic predisposition interact to promote the induction of an autoimmune response against β-cells. Viral infections have been considered the most important environmental factors in initiating the process of autoimmune destruction of β-cells.Viruses can damage pancreatic β-cells by inducing an autoimmune response against β-cells (phenomenon of molecular mimicry). In recent years, in addition to viral infections, several epidemiological studies have described new environmental factors, including vitamin D deficiency, which may play a role in triggering islet autoimmunity and T1DM in subjects at high genetic risk. That is, vitamin D deficiency may play a role in the pathogenesis of T1DM

Therefore, we understand that there is no discordance between the paragraphs of the abstract: “...about the association between vitamin D status in the pathogenesis of T1DM.” and the Introduction: “...(a) research progress on the possible function of vitamin D status as an environmental risk factor in the pathogenesis of T1DM.”

Vitamin D Synthesis and Metabolism

a) In the absorption of vitamin D, it is suggested to describe the importance of intestinal microbiota.

Although the importance of the intestinal microbiota in the absorption of vitamin D is a very interesting topic, we understand that it is not the preferred subject of this review.

b) It is necessary to enrich the non-genomic mechanisms that vitamin D exerts specifically on the antioxidant effect or on the production of nitric oxide.

As previously discussed, the antioxidant effect of vitamin D does not appear to be involved in the etiopathogenesis of T1DM.

c) In the genomic mechanisms, focus on illustrating the specific mechanism on cytokine production (Figure 1).

As mentioned above, this review is oriented to clinicians. Therefore, it is beyond the scope of this review to delve into the non-genomic mechanisms that vitamin D exerts on cytokine synthesis.

d) To deepen in the explanation of the heterodimerization phenomenon and its impact on the functionality of nuclear receptors. As described, it seems that this phenomenon is simple, however, it is more complex than one imagines. Please expand on the heterodimerization phenomenon.

Of course, this is a complex phenomenon, but we believe that it would be beyond the scope of this review to go into its molecular explanation.

Pathogenesis and Natural History of Type 1 Diabetes Mellitus

A) Avoid using repetitive premises such as the one described in the first three lines of the abstract and in lines 106-107 and 117-118.

It is inevitable that there may be some repetition between the Abstract and some sections of the article.

B) Subsequent to the following paragraph it is necessary to raise, what could represent the development of molecular experiments based on cell culture and viruses related to the pathophysiology of type 1 diabetes.

However, it is extremely difficult to prove a cause-effect relationship between viral infections and the development of T1DM, as the period between viral exposure and the onset of clinical symptoms of T1DM is often very long.

Unfortunately, our experience is exclusively clinical and we are not able to design molecular experiments in this field.

C) I think there is a disorder in how the pathogenesis of diabetes is approached:

It is suggested first to clarify that the phenomenon of autoimmunity is related both to factors proper to the organism and to environmental factors and the interaction of both.

In that order, first address the factors of the organism itself such as genetic predispositions and then continue with the environmental factors: toxins and viruses.

Next, address the interactions of both variables: nutritional factors, including (breastfeeding, early food consumption in infancy, vitamin d intake). In addition, the role of the microbiota, that may be influenced by diet and how dysbiosis, avitaminosis correlate with autoimmunity, until the end addressing epigenetics.

Part of the text has been deleted. From “Vitamin D deficiency is common in the pediatric population with T1DM....” to “...vitamin D in early life reduces the risk of diabetes” (lines 151-162 of the first version).

And now the sequence of section 3 (Pathogenesis and natural history of T1DMD) is quite similar to that advised by the reviewer:

(1) Reference is made to: “the genetic predisposition to β-cell autoimmunity in individuals with specific human leukocyte antigen (HLA-DR3-DQ2 or HLA-DR4-DQ8), and other genes related to autoimmunity (CTLA4, PTPN22, L2RA, etc.) or to β-cells (INS, GLIS3, CTSH, etc.).”

(2) Then it is said: “...the selective destruction of β-cells would be caused by an interaction between risk genes and environmental factors.” To date, viral infections have been considered the most important environmental factors in initiating the process of autoimmune destruction of β-cells (phenomenon of molecular mimicry).”

(3) And it continues: “In recent years, in addition to viral infections, several epidemiological studies have described new environmental factors (cow's milk, gluten or Omega-3 fatty acids intake, dysbiosis, vitamin D deficiency, etc.) which may play a role in triggering islet autoimmunity and T1DM in subjects at high genetic risk.”

(4) To finish with: “All of these factors could alter gene expression through epigenetic mechanisms (particularly DNA methylation, histone modifications or long non-coding RNA) thereby inducing an aberrant immune response and progressive β-cells destruction (Figure 2), and thus be involved in the pathogenesis of T1DM.”

D) In line 171, only the phenomenon of immunological tolerance is mentioned without defining the concept, it is necessary to recognize that this phenomenon is complex and is integrated by several variables.

The concept of immunological tolerance is commonly used, and we have not been forced to define it.

In short: specific absence of response of the immune system to a self-antigen. This is an active state (not a simple absence of response), endowed with specificity and memory.

E) In line 197 there is a grammatical error.

Previous text... A higher proportion of plasma cells into insulitis has been associated with a faster β-cell decline [39].”

has been changed to...”A higher proportion of plasma cells in insulitis has been associated with a faster β-cell decline [39].”

F) The information in lines 209-210 is repetitive with that previously shown in line 143.

We consider that the sentences mentioned are not redundant and that both fit in the text.

Line 146-147: “as the period between viral exposure and the onset of clinical symptoms of T1DM is often very long.”

Lines: 205-208): “ usually appear several years after the onset of the autoimmune process, when most of the pancreatic β-cells have been destroyed (stage 3), and T1DM is definitively established”

Immunomodulatory Effects of Vitamin D in Autoimmune Diseases

a) To deepen the mechanism of action of vitamin D on cytokine production.

It is beyond the scope of this article to delve into the mechanism of vitamin D on cytokine production. In this section we have tried to simplify the immunomodulatory effects of vitamin D on various immune cell lineages. In fact, a brief summary (table 1) on the immunomodulatory effects of vitamin D has been added for better understanding of the text.modulatory effects of vitamin D has been added for better understanding of the text

b) The information in lines 227-228 is somewhat repetitive with lines 236 -237.

The information on lines 233-239 has been modified.

Previous text... “Vitamin D promotes the differentiation of T-helper lymphocytes from a Th1 and Th17 profile to a Th2 and Treg profile, increasing the release of anti-inflammatory cytokines such as IL-4, IL-5 and IL10, while decreasing the production of pro-inflammatory cytokines, including IL-2, IFN-γ, TNF-α, IL-17 and IL-21.”

Has been changed to...“Vitamin D promotes the differentiation of T-helper lymphocytes from a Th1 to a Th2 phenotype. This change implies inhibition of inflammatory cytokine production (IL-2, IFN-γ, and TNF-α) and an increased production of anti-inflammatory cytokines (IL-4, IL-5, and IL-10). Furthermore, vitamin D affects differentiation to Th17 phenotype, leading to a decrease in the production of inflammatory cytokines (IL-17 and IL-21), and facilitates the induction of T reg cells with increased production of anti-inflammatory cytokines (IL-10 and TGF-β).”

Vitamin D Status and the Risk of T1D

A) To better contrast this analysis, I suggest integrating another study describing lack of differences in serum vitamin D3 levels.

Polat İ, Can Yılmaz G, Dedeoğlu Ö. Vitamin D and Nerve Conduction In Pediatric Type-1 Diabetes Mellitus. Brain Dev. 2022 May;44(5):336-342. doi: 10.1016/j.braindev.2022.01.001. Epub 2022 Jan 15. PMID: 35042650.

It seems to us a very interesting reference (their results suggest that hypovitaminosis D could lead to the development of neuropathic changes already in the early stages of the disease) that we will take into account in the evolutionary control of our patients with T1DM.

However, we believe that this specific topic is beyond the scope of this review.

B) In relation to the information in the following premise:

Specifically, vitamin D deficiency appears to be much more prevalent in patients with T1DM than in healthy individuals [13].

It is suggested to analyze and integrate the information contained in the following study:

Hou Y, Song A, Jin Y, Xia Q, Song G, Xing X. A dose-response meta-analysis between serum concentration of 25-hydroxy vitamin D and risk of type 1 diabetes mellitus. Eur J Clin Nutr. 2021 Jul;75(7):1010-1023. doi: 10.1038/s41430-020-00813-1. Epub 2020 Nov 24. PMID: 33235321; PMCID: PMC8266682.

Sorry, this sentence has been deleted due to an inadvertent typographical error.

C) In the premises described between lines 272- 274 and 276-278 there are no bibliographic citations.

References cited subsequently (Infante et al., 2019. Manousaki et al., 2021. Najjar et al., 2021. Wang et al., 2014. Habibian et al., 2019. Ran et al., 2021) suggest that vitamin D status may be an important environmental risk factor in the pathogenesis of T1DM, rather than a consequence of pathophysiological changes resulting from the disease.

D) In paragraph 287 to 289, two studies are described and the reference of only one study is cited.

This sentence only referred to a study: [51] Najjar et al. Vitamin D and Type 1 Diabetes Risk: A. Systematic Review and Meta-Analysis of Genetic Evidence. Nutrients 2021, 13, 4260.

E) In the following paragraph, there are no bibliographic references for either of the two premises. Avoid combining evidence from animal and human studies:

Experimental studies using non-obese diabetic mice as a model of human T1DM have demonstrated protective effects of vitamin D against islet autoimmunity and progressive β-cell dysfunction. Also importantly, vitamin D deficiency in early life results in a higher incidence and earlier onset of diabetes. Calcitriol and its analogues have also been shown to prevent insulitis and thus diabetes, especially when given at an early age (when the immune attack on β-cells is in its early stages).

We had placed the bibliographical references in the wrong place. In addition, a new reference has been added. Previous text has been changed to:

Experimental studies using non-obese diabetic mice as a model of human T1DM have demonstrated protective effects of vitamin D against islet autoimmunity and progressive β-cell dysfunction. Also importantly, vitamin D deficiency in early life results in a higher incidence and earlier onset of diabetes. Calcitriol and its analogues have also been shown to prevent insulitis and thus diabetes, especially when given at an early age (when the immune attack on β-cells is in its early stages) [38, 64, 65].”

We have avoided combining evidence from animal and human studies. In fact, this paragraph serves as a preamble to the next paragraph:

The efficacy of vitamin D in halting or reversing islet autoimmunity observed in preclinical studies has stimulated numerous interventional studies and randomised controlled trials that have established beneficial clinical effects of different forms of vitamin D or analogues (in addition to insulin therapy) in patients with T1DM...”

F) In table 1 it is suggested to integrate one more column with a summary of the main results of each study.

A new column has been added to Table 2 with the most significant findings in each of the interventional studies.

Table 1. Interventional studies in children/adolescents with new-onset T1DM resulting in preservation of residual pancreatic β-cell function and improved glycaemic control.

Author, Year and Country

Study design

Suplementation dosage Duration

Significants findings

Gabbay et al., 2012 (Brazil) [67]

Randomized, double blind, placebo-controlled, prospective trial

Cholecalciferol (2000 IU/d for 18 months)

Decrease in Hb1Ac levels

Decrease in autoantibody titers

Stimulated C-peptide enhancement

Increase Treg percentage

Ataie-Jafari et al., 2013 (Iran) [82]

Randomized, single blind, placebo-controlled trial

Alfacalcidol (0.25 ug/twice daiy for 6 months)

Improved stimulated C-peptide

Federico et al, 2014 (Italy) [78]

Pilot interventional study

Calcidiol (10-30 ug/d for one year)

Decrease insulin requirements

Stability of fasting C-peptide levels

Inhibition of GAD-65 antibodies

Treiber et al., 2015 (Austria) [68]

Randomized, double blind, placebo-controlled, prospective trial

Cholecalciferol (70 IU/kg/d for 12 months)

Decrease in Hb1Ac levels

Stimulated C-peptide enhancement

Reduction in daily insulin doses

Increase Treg percentage

Panjiyar et al., 2018 (India) [69]

Prospective, case-control, interventional study

Cholecalciferol (3000 IU/d for one year)

Decrease in Hb1Ac levels

Reduction in daily insulin doses

Stimulated C-peptide enhancement

Cadario et al., 2019 (Italy) [74]

Cases study

Cholecalciferol (1000 IU/d) plus EPA+DHA (50-60 mg/kg/d) for 12 months

Decrease insulin requirements

Reddy et al., 2022 (India) [75]

Pilot study

Cholecalciferol (2000 IU/d) plus lansoprazole (15-30 mg) for six months)

Decrease insulin requirements

Slower fasting peptide-C decline

Pinheiro et al., 2023 (Brasil, Italy) [76]

Case-control study

Cholecalciferol (5,000 IU/d) plus sitagliptin (50 mg/day) for 12 months

Longer duration of the remission phase

Nwosu et al., 2024 (Massachusetss, USA) [77]

Randomized, double blind, placebo-controlled, prospective trial

Ergocalciferol (50,000 IU/wk for 2 months, then fortnightly for 10 months)

Decrease insulin requirements

Reduction TNF-α concentration

G) The following paragraphs are somewhat repetitive and with contrasting ideas:

In addition, alphacalcidol improves immune senescence by acting as an anti-inflammatory agent through increased IL-10 and decreased IL6/IL-10 ratio, and also improves cellular immunity through increased CD4/CD8 ratio [81]. Few

Indeed, as mentioned above, several randomised controlled trials in the last decade have shown that vitamin D supplementa tion, especially as cholecalciferol, appears to preserve residual β-cell function and im prove glycaemic control in children and adolescents with new-onset T1DM through its

We are sorry, but we do not see any conflicting ideas between these two texts. The second text can be considered as the epilogue to section 6 (Vitamin D Supplementation in Type 1 Diabetes).

H) It is also suggested to consider the prevalence of patients with type 1 diabetes in. Where it would be difficult to measure vitamin D levels. Based on the above, what options or suggestions would you have for treating physicians in these countries?

In the bibliographic review we have found published data on this subject carried out in countries of the five continents.

As noted in section 7 (Conclusions): “Most of the researchers conclude their papers by pointing out that large-scale prospective randomized controlled trials are urgently needed to definitively establish the role of vitamin D in T1DM.” When this happens we will be in a position to suggest qualified options to clinicians in any country.

I) For the above, also, the main weaknesses of a narrative review should be recognized; alluding to the possible biases involved in the classification of the studies considered, trying to have solid arguments to minimize the influence of these bias factors.

Narrative reviews are ideal for presenting general aspects of a given topic, but conceptually they carry a great weakness (subjectivity and/or unclear methodology).

To reduce bias, many more articles have been analyzed than those listed as references.

J) The last conclusion is inappropriate, since it is a suggestion and does not agree with the main objective.

Surprisingly, in the bibliographic review carried out, no author has hinted at this possibility which, on the other hand, we feel obliged to point out. Moreover, we are proud to be able to do so on the basis of an extensive bibliographic analysis:

We believe that the fact that interventional studies generally have better results in patients newly diagnosed with diabetes should not be surprising. This should lead us to believe that the weakness of the aforementioned interventional studies does not lie in the heterogeneity of their characteristics, but rather in their relationship with functional or residual β-cell mass. To date, all intervention studies have been carried out in the clinical phase of T1DM, whereas it would be desirable to be able to do so in the early stages of the autoimmune process (pre-diabetes). In other words, the efforts of the desired randomised controlled trials should also be directed towards obtaining biomarkers that can detect the onset of autoimmune insulitis and, in these precise circumstances, initiate vitamin D or analogues supplementation in addition to insulin treatment. This could possibly slow down or prevent the autoimmune process from continuing its progressive course and, as far as possible, avoid reaching the clinical and irreversible phase of the disease.”

K) The algorithm in the form of a diagram is inappropriate, if the criteria are not standardized in a larger number of studies and the unfeasibility of these criteria in underdeveloped countries is not recognized.

Completely in agreement. We have deleted the algorithm, but kept the explanatory text. Most of the researchers conclude their papers by pointing out that large-scale prospective randomized controlled trials are urgently needed to definitively establish the role of vitamin D in T1DM. In the meantime, we would suggest a protocol for vitamin D supplementation in children and adolescents with new-onset T1DM based on the data in this narrative review

We would like to express our thanks to referee for your suggestions and positive criticisms.

We hope every made question have been answered adequately.

Yours sincerely,

Reviewer 2 Report

Comments and Suggestions for Authors

Summary

This narrative review summarizes the aspects of the association between vitamin D status in the pathogenesis of type 1 diabetes mellitus (T1DM) and the potential role of vitamin D supplementation in the prevention and treatment of T1DM. This review paper includes two main aims, highlighting the (a) research progress on the possible function of vitamin D status as an environmental risk factor in the pathogenesis of T1DM and (b) the assessment of the potential role of vitamin D in the prevention and treatment of T1DM. The literature, spanning over a decade, search strategy is electronic searching of the PubMed. However, a few further suggestions, to improve the quality of this manuscript, would be important to support the depth and rigor of the conclusions drawn and would be helpful for the learn and guide the research between vitamin D and T1DM.

Specific comments

  1. In the abstract of this review paper, the authors argue that, “…..there does not appear to be an association between vitamin D status before diagnosis and the onset of T1DM later in life”, at the same time, in the conclusion, they point “….. vitamin D deficiency in early life results in a higher incidence and earlier onset of diabetes”. For the clarity purposes of the wide reader community, authors should carefully look at these statements and may wish revise as appropriately drawing a distinctive separation between “vitamin D status” “vitamin D deficiency”. For instance, they could describe when the cases are when “vitamin D status” is not a “vitamin D deficiency” or vice vera.
  2. While the main view, in the development of type 1 diabetes mellitus (T1DM), is that autoimmune mechanisms (aka autoreactive T cells) mistakenly destroy healthy β-cells. Most recently, an alternative view such that the dysfunction could occur in β-cell and β-cell biosynthetic stress (e.g., metabolic stress) could provoke the immune attack is gaining momentum. Within this context, in section 6. “Vitamin D Supplementation in Type 1 Diabetes”, while describing preventive role of vitamin D in T1DM, the abrupt switching between two important viewpoints of development of T1DM limits the unique of the manuscript and is difficult to follow. Alternatively, keeping in view that the dysfunction could occur both in the β-cell and immune system, I hope the authors should include a separate discussion to strengthen the aim (b) of this review paper “the assessment of the potential role of vitamin D in the prevention and treatment of T1DM” highlighting how vitamin D based (immune-)therapy, aimed at revitalizing the β-cell centric dysfunction, would best delays/prevents the T1DM. The authors could consider carefully revising the section 6, and relevantly, including a new section in the manuscript (ideally after section 6) with briefly introducing a refence to the role of the β-cell, as a key contributor to the disease, in their own demise.
  3. As for section 4 “Immunomodulatory Effects of Vitamin D in Autoimmune Diseases”, the authors could also consider providing a brief summary of the findings/meaning of a group of studies reviewed here before moving to the next series of section(s) to better allow readers to follow the justification and rationale for the work. This would be especially helpful for readers who may approach this work from a background only in vitamin D biology or instead in islet biology but not experienced in the other field.

Minor comments

  1. Line 17 and 18: Abbreviation “T1DMD” either must be defined at first instance or corrected accordingly if authors are referring to type 1 diabetes mellitus (T1DM).
  2. Line 28-30, and 32-33: “Type 1 diabetes mellitus (T1DM) ……. it can affect people of any age”, “Interestingly, ……increased significantly in recent decades” missing references and must be supported with appropriate recent references.
  3. Table 1 summarizes, as the title indicates, vitamin D supplementation interventional studies in children/adolescents with new-onset T1DM, it lacks a brief summary of the major findings/parameters suggesting the preservation of residual pancreatic β-cell function and improved glycaemic control.

Author Response

Responses to reviewer-2

First of all, we would like to thank you for your suggestions as well as your words of encouragement regarding this article.

First of all, we would like to thank you for your advice regarding this article

NOTE: The corrected text of the new version is in red

Comments and Suggestions for Authors

Summary

This narrative review summarizes the aspects of the association between vitamin D status in the pathogenesis of type 1 diabetes mellitus (T1DM) and the potential role of vitamin D supplementation in the prevention and treatment of T1DM. This review paper includes two main aims, highlighting the (a) research progress on the possible function of vitamin D status as an environmental risk factor in the pathogenesis of T1DM and (b) the assessment of the potential role of vitamin D in the prevention and treatment of T1DM. The literature, spanning over a decade, search strategy is electronic searching of the PubMed. However, a few further suggestions, to improve the quality of this manuscript, would be important to support the depth and rigor of the conclusions drawn and would be helpful for the learn and guide the research between vitamin D and T1DM.

Specific comments

1. In the abstract of this review paper, the authors argue that, “…..there does not appear to be an association between vitamin D status before diagnosis and the onset of T1DM later in life”, at the same time, in the conclusion, they point “….. vitamin D deficiency in early life results in a higher incidence and earlier onset of diabetes”. For the clarity purposes of the wide reader community, authors should carefully look at these statements and may wish revise as appropriately drawing a distinctive separation between “vitamin D status” “vitamin D deficiency”. For instance, they could describe when the cases are when “vitamin D status” is not a “vitamin D deficiency” or vice vera.

Both in the abstract (lines 17-18: ...there does not appear to be an association between pre-diagnosis vitamin D status and the occurrence of T1DM later in life.) and in the section Vitamin D Status and the Risk of T1DM (lines 355-356: Results from long-term follow-up studies in children suggest no association between pre-diagnosis vitamin D status and the occurrence of T1DM later in life. ) and in the conclusions (line 502-504: ...results from long-term follow-up studies in children suggest that there is no association between vitamin D status before diagnosis and the occurrence of T1DM later in life) express the same idea. The articles by Raab et al. (ref. 61), Simpson et al. (ref. 62) and Mäkinn et al. (ref. 63) suggest that the development of T1DM would not be associated with vitamin D status.

The alluded sentence would correspond to lines 161-162 of the first version: “There is also high-level evidence (systematic reviews, meta-analyses) that adequate vitamin D in early life reduces the risk of diabetes.” This sentence has been deleted as it was due to a typographical error.

2. While the main view, in the development of type 1 diabetes mellitus (T1DM), is that autoimmune mechanisms (aka autoreactive T cells) mistakenly destroy healthy β-cells. Most recently, an alternative view such that the dysfunction could occur in β-cell and β-cell biosynthetic stress (e.g., metabolic stress) could provoke the immune attack is gaining momentum. Within this context, in section 6. “Vitamin D Supplementation in Type 1 Diabetes”, while describing preventive role of vitamin D in T1DM, the abrupt switching between two important viewpoints of development of T1DM limits the unique of the manuscript and is difficult to follow. Alternatively, keeping in view that the dysfunction could occur both in the β-cell and immune system, I hope the authors should include a separate discussion to strengthen the aim (b) of this review paper “the assessment of the potential role of vitamin D in the prevention and treatment of T1DM” highlighting how vitamin D based (immune-)therapy, aimed at revitalizing the β-cell centric dysfunction, would best delays/prevents the T1DM. The authors could consider carefully revising the section 6, and relevantly, including a new section in the manuscript (ideally after section 6) with briefly introducing a refence to the role of the β-cell, as a key contributor to the disease, in their own demise.

On line 381 there was a typographical error in the text which may have made it difficult to understand (dysfunction vs. destruction).

Previous text... “...against islet autoimmunity and progressive β-cell dysfunction.”

has been changed to... “...against islet autoimmunity and progressive β-cell destruction.”

The etiopathogenesis of type 1 and type 2 diabetes are different.

T2DM (approximately 90% of all cases of diabetes) is a chronic metabolic disorder characterized by insulin resistance and/or and beta-cell dysfunction and consequentially high blood glucose. Under hyperglycaemic conditions, reactive oxygen species (hydrogen peroxide and superoxide anion) are likely to be involved in pancreatic beta-cell dysfunction and insulin resistance. Hyperglycaemia-induced oxidative stress would play a substantial role in the complications of this disease. Indeed, there is increasing awareness and evidence that diabetes mellitus, particularly type 2 diabetes, is significantly modulated by oxidative stress.

There is a recent narrative review that is very illustrative on this issue: Singh, et al. Mechanistic Insight into Oxidative Stress-Triggered Signaling Pathways and Type 2 Diabetes. Molecules 2022, 27, 950.

On the other hand, T1DM -subject of this review- is induced by destruction of pancreatic beta-cells which is mediated by an autoimmune mechanism and consequent inflammatory process. Evidently, poorly controlled T1DM will cause oxidative stress and likely contribute to the development of subsequent vascular complications. However, due to the low functional islet mass at the onset of the disease, it does not appear to substantially contribute to the functional impairment of pancreatic beta cells.

3. As for section 4 “Immunomodulatory Effects of Vitamin D in Autoimmune Diseases”, the authors could also consider providing a brief summary of the findings/meaning of a group of studies reviewed here before moving to the next series of section(s) to better allow readers to follow the justification and rationale for the work. This would be especially helpful for readers who may approach this work from a background only in vitamin D biology or instead in islet biology but not experienced in the other field.

A brief summary (table 1) on the immunomodulatory effects of vitamin D has been added for better understanding of the text (lines: 230-231):

Table 1. Immunomodulatory effects of vitamin D on immune cells.

Immune Cell Type

Vitamin D- Induced Effect

Macrophages

Pro-inflammatory IL-1, IL-6, IL-8, IL-12

Anti-inflammatory IL-10

Natural Killer cells

Pro-inflammatory IFN-γ

Anti-inflammatory IL-4

Dendritic cells

Pro-inflammatory IL-2, IL-6, Il-12

Anti-inflammatory IL-10

DCs differentiation (tolerogenic DCs)

Antigen presentating cells (T-cells anergy)

CD4+ T cells

Hyperactivation

Th1, Th17

Th2, Treg

Anti-inflammatory IL-4, IL-5, IL-10, TGF-β

Pro-inflammatory IL-2, IFN-γ, TNF-α, IL,17 IL-21

CD8+ T cells

Hyperactivation

B cells

B cells proliferation and differentiation into plasma cells

Memory B cells formation

Autoantibody production

Ref.: Park et al., 2004 [5]. Infante el al., 2019 [38]. He et al., 2022 [39]. Prietl et al., 2013 [42]. Dankers et al., 2017 [43]. Rack, et al., 2018 [44]. Cyprian, et al., 2019 [45]. Gallo et al, 2023 [46]. Ghaseminejad-Raeini et al., 2023 [47]. Galdo-Torres et al , 2025 [48]

Minor comments

1. Line 17 and 18: Abbreviation “T1DMD” either must be defined at first instance or corrected accordingly if authors are referring to type 1 diabetes mellitus (T1DM).

It is a typographical error (Abstract).

Previous text... T1DMD…has been changed to... T1DM (line: 18)

2. Line 28-30, and 32-33: “Type 1 diabetes mellitus (T1DM) ……. it can affect people of any age”, “Interestingly, ……increased significantly in recent decades” missing references and must be supported with appropriate recent references.

There was a mismatch between the text and the corresponding reference, which has been modified.

Previous text... “Interestingly, the incidence of T1DM has increased significantly in recent decades.”

has been changed to... “Interestingly, the incidence of T1DM has increased significantly in recent decades [2].

[2] Patterson, et al. Diabetologia. 2019, 62, 408–417.

Table 1 summarizes, as the title indicates, vitamin D supplementation interventional studies in children/adolescents with new-onset T1DM, it lacks a brief summary of the major findings/parameters suggesting the preservation of residual pancreatic β-cell function and improved glycaemic control.

A new column has been added to Table 2 with the most significant findings in each of the interventional studies.

Table 1. Interventional studies in children/adolescents with new-onset T1DM resulting in preservation of residual pancreatic β-cell function and improved glycaemic control.

Author, Year and Country

Study design

Suplementation dosage Duration

Significants findings

Gabbay et al., 2012 (Brazil) [67]

Randomized, double blind, placebo-controlled, prospective trial

Cholecalciferol (2000 IU/d for 18 months)

Decrease in Hb1Ac levels

Decrease in autoantibody titers

Stimulated C-peptide enhancement

Increase Treg percentage

Ataie-Jafari et al., 2013 (Iran) [82]

Randomized, single blind, placebo-controlled trial

Alfacalcidol (0.25 ug/twice daiy for 6 months)

Improved stimulated C-peptide

Federico et al, 2014 (Italy) [78]

Pilot interventional study

Calcidiol (10-30 ug/d for one year)

Decrease insulin requirements

Stability of fasting C-peptide levels

Inhibition of GAD-65 antibodies

Treiber et al., 2015 (Austria) [68]

Randomized, double blind, placebo-controlled, prospective trial

Cholecalciferol (70 IU/kg/d for 12 months)

Decrease in Hb1Ac levels

Stimulated C-peptide enhancement

Reduction in daily insulin doses

Increase Treg percentage

Panjiyar et al., 2018 (India) [69]

Prospective, case-control, interventional study

Cholecalciferol (3000 IU/d for one year)

Decrease in Hb1Ac levels

Reduction in daily insulin doses

Stimulated C-peptide enhancement

Cadario et al., 2019 (Italy) [74]

Cases study

Cholecalciferol (1000 IU/d) plus EPA+DHA (50-60 mg/kg/d) for 12 months

Decrease insulin requirements

Reddy et al., 2022 (India) [75]

Pilot study

Cholecalciferol (2000 IU/d) plus lansoprazole (15-30 mg) for six months)

Decrease insulin requirements

Slower fasting peptide-C decline

Pinheiro et al., 2023 (Brasil, Italy) [76]

Case-control study

Cholecalciferol (5,000 IU/d) plus sitagliptin (50 mg/day) for 12 months

Longer duration of the remission phase

Nwosu et al., 2024 (Massachusetss, USA) [77]

Randomized, double blind, placebo-controlled, prospective trial

Ergocalciferol (50,000 IU/wk for 2 months, then fortnightly for 10 months)

Decrease insulin requirements

Reduction TNF-α concentration

We would like to express our thanks to referee for your suggestions and positive criticisms.

We hope every made question have been answered adequately.

Yours sincerely,

Round 2

Reviewer 1 Report

Comments and Suggestions for Authors

Three things that I disagree with and need to be rechecked.

  1. It is necessary to integrate a paragraph that analyzes what is novel and different that this review contributes compared to other previous similar reviews, this with the purpose that the reader realizes the evolution of the knowledge that has been had on this area of knowledge.
  2. It is convenient and necessary to explain in a general way some variables that were not addressed: oxidative stress which, although it does not participate in the origin of the disease, there is previous research that addresses the importance of this factor in the development of the disease, which is ultimately part of the pathogenesis. It is necessary to add the concept of immunological tolerance which in some cases is not so common for the various health professionals. It is necessary to include the importance of the intestinal microbiota, since this information does not disorient the subject of study, on the contrary, it complements and enriches it.
  3. The conclusions are still broad and need to be summarized.

Author Response

Responses to reviewer-1 (second round)

First of all, we would like to thank you for your suggestions regarding this article.

NOTE: The corrected text of the new version is in red

Comments and Suggestions for Authors

Three things that I disagree with and need to be rechecked.

  1. It is necessary to integrate a paragraph that analyzes what is novel and different that this review contributes compared to other previous similar reviews, this with the purpose that the reader realizes the evolution of the knowledge that has been had on this area of knowledge.

The objectives of this narrative review -we are aware of its implicit limitations- have been to analyze the research progress on the possible function of vitamin D status in the pathogenesis of T1DM and the assessment of the potential role of vitamin D in the prevention and treatment of T1DM. The data accumulated in this regard over the last thirteen years have been presented, with their "lights and shadows"...

The third paragraph of the Conclusions has been modified to indicate what is novel and different about this review (lines: 587-588):

Previous text...

We believe that the fact that interventional studies generally have better results in patients newly diagnosed with diabetes should not be surprising. This should lead us to believe that the weakness of the aforementioned interventional studies does not lie in the heterogeneity of their characteristics, but rather in their relationship with functional or residual β-cell mass.,,”

has been changed to…

"We believe that the fact that interventionalist studies obtain, in general, better results in patients recently diagnosed with diabetes should not be surprising. What is novel and different about this review is to stimulate us to think that the weakness of the aforementioned interventional studies would not lie in the heterogeneity of their characteristics, but rather in their relationship with functional β-cell mass or residual β-cell mass." ...."

  1. It is convenient and necessary to explain in a general way some variables that were not addressed: oxidative stress which, although it does not participate in the origin of the disease, there is previous research that addresses the importance of this factor in the development of the disease, which is ultimately part of the pathogenesis. It is necessary to add the concept of immunological tolerance which in some cases is not so common for the various health professionals. It is necessary to include the importance of the intestinal microbiota, since this information does not disorient the subject of study, on the contrary, it complements and enriches it.

Regarding the role of oxidative stress, the following paragraph has been added to section “Pathogenesis of T1DM” (lines: 241-258):

Reactive oxygen species (ROS) include several oxygen-containing free radicals, such as superoxide anion, hydroxyl radical and hydrogen peroxide. Excessive amounts of ROS may cause deleterious oxidative damage to biomolecules (DNA, proteins and lipids), consequently leading to cell death. An imbalance between the production and accumulation of ROS and enzimatic antioxidant systems, including glutathione peroxidase, superoxide dismutase, and catalase, is known as “oxidative stress”. The main source of free radicals responsible for oxidative stress is mitochondrial respiration. but they can originate from exogenous sources, such as ageing, inflammation, radiation, and toxic chemicals. That is, chronic inflammatory processes would aggravate oxidative stress and enhance ROS generation [45, 46]. Although hyperglycemia is a key factor of oxidative stress, the interaction of autoimmune and inflammatory pathways in T1DM, independent of glucose levels, could contribute to the overproduction of ROS. These ROS would cause oxidative damage to cellular structures leading to β-cell death and disease progression. In fact, mitochondria-derived free radicals have been shown to contribute to the immune-mediated β-cell destruction process, either through induction of toxicity by cytokines (IL-1, TNF-α and IFN-γ) or due to low expression of antioxidant enzymes in islet [47-49].”

On the concept of Immune tolerance, the following note has been added (lines: 185-190):

(Note: Immune tolerance is a physiologic state where the immune system does not react against the body's own cells and tissues, preventing an immune attack against self-antigens. This is an active state (not a simple absence of response), endowed with specificity and memory. Autoimmune diseases arise when this tolerance is disrupted, leading to the immune system attacking the body's own tissues)

In relation to the role of gut microbiota dysbiosis in T1DM pathophysiology the following text has been modified (lines 162-173)

Previous text...

Dysbiosis is associated with abnormalities in the activity of the immune system. In fact, lower gut microbiota diversity appears to be associated with an increased risk of T1DM (33)”

has been changed to…

A complex correlation between gut microbiota, the immune system and intestinal permeability has been identified, although it has not been fully unraveled. Several cross-sectional studies have found large significant differences in microbiota composition between subjects with T1DM or islet autoimmunity and healthy controls. Hypothetically, intestinal dysbiosis would lead to dysregulation of the immune response, including both the innate and adaptive immune systems, and/or intestinal permeability, ultimately leading to β-cell destruction and the onset of T1DM in genetically susceptible individuals. Increased intestinal permeability would allow antigens and pathogens to cross the intestinal barrier, which could trigger or exacerbate immune responses against pancreatic beta cells, contributing to the etiopathogenesis of T1DM. However, the exact role of the intestinal microbiota in the pathogenesis of T1DM remains controversial (33-36)”

  1. The conclusions are still broad and need to be summarized.

The last paragraph of the conclusions seems to us more appropriate to place it at the end of the section “Vitamin D Supplementation in Type 1 Diabetes Mellitus” (lines: 542-563).. This would also reduce the length of the Conclusions:

To conclude, we would suggest a protocol for vitamin D supplementation in children and adolescents with new-onset T1DM based on the data in this narrative review. Obviously, serum calcidiol levels would be measured in all newly diagnosed T1DM patients. If calcidiol levels are less than 30 ng/ml, they should receive cholecalciferol supplementation (1000 to 2000 IU/day) to maintain serum calcidiol concentration between 30-50 ng/ml (previously defined as optimal vitamin D levels). Patients with serum calcidiol concentrations above 30 ng/mL at the onset of diabetes should be monitored with serial calcidiol concentrations and cholecalciferol supplementation should be initiated if serum calcidiol concentrations are below 30 ng/mL. Cholecalciferol supplementation should be continued for at least one year to ensure optimal vitamin D benefit. Of course, residual β-cell function (fasting plasma C-peptide leves), glycaemic control (HbA1c levels and/or fasting plasma glucose), T1DM-associated autoantibodies (islet autoantibodies) and exogenous insulin requirements from diagnosis should be monitored periodically, together with vitamin D (calcidiol) levels.”

We would like to express our thanks to referee for your suggestions and positive criticisms.

We hope every made question have been answered adequately.

Yours sincerely,

Reviewer 2 Report

Comments and Suggestions for Authors

The authors have adequately addressed the comments with appropriate corrections as well as by adding new texts/tables(s). There is a small font mistake: Line 241, "autoanti-body" where "auto" is in bold font.

Author Response

Responses to reviewer-2 (second round)

First of all, we would like to thank you for your suggestions regarding this article.

NOTE: The corrected text of the new version is in red

Comments and Suggestions for Authors

The authors have adequately addressed the comments with appropriate corrections as well as by adding new texts/tables(s). There is a small font mistake: Line 241, "autoanti-body" where "auto" is in bold font.

The typographical error (line 294) has been corrected. Thank you very much.

Yours sincerely,